

# Heat stress risk in European dairy cattle husbandry under different climate change scenarios - uncertainties and potential impacts

Sabrina Hempel[1], Christoph Menz[2], Severino Pinto[1], Elena Galán[3], David Janke[1], Fernando Estellés[4], Theresa Müschner-Siemens[1], Xiaoshuai Wang[5], Julia Heinicke[1], Guoqiang Zhang[5], Barbara Amon[1], Agustín del Prado[3,6], and Thomas Amon[1,7]

[1]Leibniz Institute for Agricultural Engineering and Bioeconomy (ATB), Max-Eyth-Allee 100, 14469 Potsdam, Germany
[2]Potsdam Institute for Climate Impact Research (PIK), Telegraphenberg A 31, 14473 Potsdam, Germany
[3]Basque Centre for Climate Change (BC3), Sede Building 1, 1st floor, Scientific Campus of the University of the Basque Country, 48940 Leioa, Spain
[4]Institute of Animal Science and Technology, Universitat Politècnica de València, (UPV), Camino de Vera, s/n 46022 Valencia, Spain
[5]Aarhus University (AU), Department of Engineering, Blichers Allé 20, P.O. Box 50, 8830 Tjele, Denmark
[6]Basque Center for Applied Mathematics (3BCAM), Alameda de Mazarredo 14, 48009 Bilbao, Bizkaia
[7]Free University Berlin (FUB), Department of Veterinary Medicine, Institute of Animal Hygiene and Environmental Health

**Correspondence:** Sabrina Hempel (shempel@atb-potsdam.de)

**Abstract.** In the last decades, an exceptional global warming trend was observed. Along with the temperature increase, modifications in the humidity and wind regime amplify the regional and local impacts on livestock husbandry. Direct impacts include the occurrence of climatic stress conditions. In Europe, cows are economically highly relevant and are mainly kept in naturally ventilated buildings that are most susceptible to climate change. The high-yielding cows are particularly vulnerable to heat

stress. Modifications in housing management are the main measures taken to improve the ability of livestock to cope with these conditions. Measures are, however, typically taken in direct reaction to uncomfortable conditions instead of in anticipation of a long term risk for climatic stress. Moreover, measures that balance welfare, environmental and economic issues are barely investigated in the context of climate change and are thus almost not available for commercial farms. Quantitative analysis of the climate change impacts on the animal welfare and linked economic and environmental factors are rare.

Therefore, we used a numerical modeling approach to estimate the future heat stress risk in such dairy cattle husbandry systems. The indoor climate was monitored inside three reference barns in Central Europe and in the Mediterranean region. An artificial neuronal network (ANN) was trained to relate the outdoor weather conditions provided by official meteorological weather stations to the measured indoor microclimate. Subsequently, this ANN model was driven by an ensemble of regional climate model projections with three different greenhouse gas concentration scenarios. For the evaluation of the heat stress risk,

we considered the amount and duration of heat stress events. Based on the changes of the heat stress events various economic and environmental impacts were estimated.

We found that the impacts of the projected increase of heat stress risk vary dependent on the region respectively the barn, the climate model and the assumed greenhouse gas concentration. There was an overall increasing trend in number and duration of heat stress events. At the end of the century, the number of annual stress events can be expected to increase by up to 2000 hours



while the average duration of the events increases by up to 22h compared to the end of the last century. This implies strong impacts on economics, environment and animal welfare and an urgent need for mid-term adaptation strategies. We anticipated that up to one tenth of all hours of a year respectively one third of all days will be classified as critical heat stress conditions. Due to heat stress, milk yield may decrease by about 3.5% relative to the present European milk yield and farmers may expect

financial losses in the summer season of about 6.6% of their monthly income. In addition, an increasing demand for emission reduction measures must be expected, as an emission increase of about 16 Gg ammonia and 0.1 Gg methane per year can be expected under the anticipated heat stress conditions. The cattle respiration rate increases by up to 60% and the standing time may be prolonged by 1 h. This promotes health issues and increases the probability of medical treatments.

The various impacts imply feedback loops in the climate system which are presently underexplored. Hence, future in-depth

studies on the different impacts and adaptation options at different stress levels are highly recommended.

# 1 Introduction

In the last decades, a continuation of the long-term global warming trend was observed and regional and local impacts have already become apparent (WMO, 2018). These impacts are expected to become worse with ongoing climate change (Christensen

et al., 2007; van Oldenborgh et al., 2013). For Europe, temperature increase is projected in all seasons (Kjellström et al., 2018). Regional climate models anticipate a strong warming in large parts of North-East Europe that is particularly pronounced in winter. The strongest warming in summer was observed in South and South-West Europe. Along with the temperature increase, modifications in the humidity (precipitation) and wind regime are expected.

Seasonal shifts and changes in frequency and intensity of weather extremes will amplify the impacts in many economic

sectors such as agriculture (Nardone et al., 2010). It is expected that approximately 26% of all damages and losses associated with medium to large scale climate-related disaster are attributed to agriculture with its sectors crops, livestock, fisheries, aquaculture and forestry (Food and Agriculture Organization of the United Nations (FAO), 2017). So far, many studies of climate change impacts on agricultural production focus mainly on land use or crop yields (Olesen and Bindi, 2002; Kurukulasuriya and Rosenthal, 2013). Impacts on livestock are considered minor compared to crops due to more efficient, more direct and

faster adaptation strategies. For example, during hot periods grazing livestock can look for shading or livestock in barns can gather close to fans. Mechanisms of climatic effects on plants have been already implemented in numeric models decades ago such as EPIC (Erosion-Productivity Impact Calculator) plant growth model or WOFOST (WOrld FOod STudies) model (Williams et al., 1989; Diepen et al., 1989). The development of models for the livestock sector emerged in recent years and focused on field and farm scale models that map the interactions between farm components such as livestock, grassland, animal

housing, manure storage and farm management (Hutchings et al., 1996; Del Prado et al., 2006). In consequence, initiatives like



AgMIP (www.agmip.org), ISIMIP (www.isimip.org) or MACSUR (www.macsur.eu) have many more contributions in the crop sector as compared to the livestock sector.

The topic of direct climate change impacts on livestock production is becoming more and more important due to the potential consequences of climatic stress (Vitt et al., 2017). Uncomfortable climatic conditions impair animal growth, meat and milk yield and quality, egg yield, weight and quality, reproductive performance, metabolic and health status and immune response of farm animals (Nardone et al., 2010; Brouček et al., 1991; Angrecka et al., 2015). The term climatic stress (i.e., heat stress and cold stress) refers to any change to the body of livestock when trying to adapt to changing meteorological conditions. This includes physiological and behavioral changes (Galán et al., 2018). It can be caused by any combination of air movement, temperature, humidity and radiant heat (Mader et al., 2006).

Breeding is one possibility to reduce the impacts of climatic stress (Hammami et al., 2014). However, as climate change is a slow process, current measures for climate change adaptation and mitigation play only a minor role in breeding compared to the traditional economic drivers and there are contradictory aims (i.e. low heat stress susceptibility versus high yields) (Hoffmann, 2010). Hence, modifications in housing management are the main measures taken to improve the ability of livestock to cope with climatic stress conditions. Measures and systems for early warning and automatic adaptation that balance welfare, environmental and economic issues are, however, barely investigated in the context of climate change and are thus almost not available for commercial farms. In order to address this crucial and complex topic inter- and transdisciplinary research is required, incorporating natural sciences, social sciences and engineering.

According to STATISTICA (www.statista.com), approximately 47 million tons of fresh dairy products are consumed annually in the European Union. In 2016, according to EUROSTAT (ec.europa.eu/eurostat/statistics-explained), 168 million tons of milk were produced in the EU-28, nearly 97% of which were from cattle. The large-scale farming of cattle is a hot topic in public discussions related to animal welfare and emissions (Steinfeld et al., 2013). High-yielding dairy cattle have a relatively narrow range of environmental conditions for optimal milk yield and milk quality (West, 2003; Kadzere et al., 2002). In this so-called thermo-neutral zone (typically around 10°C) the cattle do not suffer significantly from climatic stress (i.e., minimal physiological effort for adaptation) which has an added value on animal welfare and health. The thermal optimum is associated with minimal methane emissions (Hempel et al., 2016b). Depending on cow individual factors, such as breed, age or productivity (milk yield), and the local environment to which the cows are adapted, the edges of the thermo-neutral zone and the stress threshold differ (e.g., the optimum is considered to be at 5°C or at 15°C) (Hahn, 1999; Kadzere et al., 2002; West, 2003; Brügemann et al., 2012; Heinicke et al., 2018). Potential stress indicators are changes in body temperature, respiration rate, milk yield, rumination activity or lying, feeding and drinking behavior (Hempel et al., 2016a; Polsky and von Keyserlingk, 2017; Curtis et al., 2017; Heinicke et al., 2018). Adaptation and recovery phases are crucial (Dinar and Mendelsohn, 2011).

The high-yielding dairy cattle are particularly susceptible to heat stress. Hence, farmers are aware of the importance and benefits of a good ventilation system for removing excess moisture (about 600 g/h per cow) and heat (about 1500 W per cow) produced by the cows in order to minimize heat stress (Pedersen and Sällvik, 2002). There are in principle two options to achieve this, mechanical and natural ventilation, of which the latter is most typical for dairy cows across Europe, as well as in many other parts of the world (Queiroz et al., 2005; Samer et al., 2011). Despite of some regional differences, all naturally



ventilated barns have the energy saving aspect in common since they do not require energy to constantly operate fans. However, they are most vulnerable to increased climate variability associated with climate change, since there is lack of precise control of the air flow. A suitable location of the building with respect to prevailing winds and surrounding trees, structures and land formations is essential.

All together this renders adaptation of dairy cattle husbandry to climate change particularly challenging and leads to various impacts, not only on animal welfare, but also on economics and environment and in the end potentially also human health. The current design of naturally ventilated barns offers only limited regulation options which have been developed to fit to the local outdoor climate in specific regions. Adaptation involves mainly short-term strategies such as turning fans or sprinklers on or off depending on the predicted outdoor temperature. A sound prediction of the anticipated number and duration of heat stress

events in naturally ventilated barns will be valuable for the farmers to schedule mid-term and optimize short-term adaptation strategies. Indoor climate modeling based on indoor measurements (together with knowledge of the range of uncertainties) can improve the assessment of future heat stress events and thus promote adaption of the husbandry system.

The interdisciplinary European project OptiBarn (www.optibarn.eu, Hempel et al. (2017a, b, c)) was designed to investigate adaptation needs and options for optimized animal-specific housing of European livestock under climate change. A modeling

system was established in the project to link measurements and modeling of barn climate (natural sciences and engineering) and research on climate-induced behavioral and physiological changes at the barn scale (veterinary and agricultural sciences) with research on climate change and economic impacts at the farm scale. Important aspects in this context are the physiological needs of the roomed livestock species as well as the regionally typical specifications of the housing. In OptiBarn, dairy cows were selected for case studies as in Europe they are economically the most relevant livestock species.

Within the OptiBarn project, meteorological data was collected inside naturally ventilated barns together with physiological and behavioral data focusing on dairy cattle farming in three reference barns in Central Europe and in the Mediterranean region in order to develop region-specific, sustainable adaptation strategies for dairy housing. This data set was used in our study, to investigate changes in the heat stress risk of dairy cattle housed in naturally ventilated barns. We hypothesize that the probability of the occurrence of critical indoor conditions depends on the barn concept and the outdoor climate conditions. We analyzed the

consequences for future heat stress risk when considering different climate projections for different regions including the effect of air movement. We discuss uncertainties of those heat stress risk projections and provide an overview of potential impacts that need further research in the future. In particular, using the database of the OptiBarn project and contemporary literature we deduce the order of magnitude of impacts on milk yield and subsequent farm income, ammonia and methane emissions, as well as respiration and activity of cows (to evaluate the impact on animal welfare and health). Eventually, we outline potential

mid-term and short-term adaptation strategies.

## 2    Data and Methods

We based our analysis on data collected within several measurement campaigns in three barns conducted during the OptiBarn project. We developed statistical models to relate the outdoor weather conditions to the indoor microclimate. The latter was





**Table 1.** Overview of on-farm measurement campaigns. Measurements were conducted approximately 3 m above the floor of the barns. The horizontal distribution of measurement points is sketched in Fig. 1 (cf. IDs). Device specifications: Comark Diligence EV N2003 sensors (Comark Limited, Hertfordshire, UK) logged temperature and relative humidity every 10 minutes (instantaneous value for the second). EasyLog USB 2+ sensors (Lascar Electronics Inc., USA) logged temperature and relative humidity every 5 minutes (instantaneous value, shortest logging rate ten seconds). 3-axis ultrasonic anemometers of type Wind Master (Gill Instruments Limited, Hampshire, UK) logged air velocity every second.

| Focus region | reference barn | Begin | End | Devices | ID |
|---|---|---|---|---|---|
| Central European maritime | Dummerstorf | 27-05-2015 | 01-11-2016 | 4 Comark Diligence EV N2003 | DT_T1 |
| Central European maritime | Dummerstorf | 01-11-2016 | 28-08-2017 | 4 EasyLog USB 2+ | DT_T2 |
| Central European maritime | Dummerstorf | 23-03-2015 | 28-08-2017 | 9 Wind Master | DT_V1 |
| Central European maritime | Dummerstorf | 23-03-2015 | 12-10-2016 | 4 additional Wind Master | DT_V2 |
| Central European maritime | Dummerstorf | 26-10-2016 | 28-08-2017 | 4 additional Wind Master | DT_V3 |
| Central European continental | Groß Kreutz | 02-06-2015 | 19-05-2017 | 8 EasyLog USB 2+ | GK_T |
| Central European continental | Groß Kreutz | 02-06-2015 | 19-05-2017 | 8 Wind Master | GK_V |
| Mediterranean | Bétera | 30-06-2016 | 06-07-2016 | 4 EasyLog USB 2+ | BT_T1 |
| Mediterranean | Bétera | 18-07-2017 | 08-09-2017 | 4 EasyLog USB 2+ | BT_T2 |

related to the stress perceived by the cows using two empirical models. An ensemble of simulations from different regional climate models (RCMs) was considered to evaluate the future heat stress risk. Anticipated impacts and adaptation options were further discussed taking into account data of the animals' physiological state and behavior collected on different farms in the OptiBarn project and in contemporary literature.

## 2.1 On-farm measurements

Our analysis was conducted based on data from three locations in Europe: two barns in Germany and one barn in Spain. The meteorological indoor data sets covered air temperature and relative humidity as well as air velocity collected between summer 2015 and summer 2017.

### 2.1.1 Reference barn Dummerstorf - Central European maritime region

The naturally ventilated dairy building is located in North-East Germany close to the Baltic Sea (Gut Dummerstorf in Mecklenburg - Western Pomerania, 42 m above sea level, cf. (Hempel et al., 2018)). It is approx. 96 m long and 34 m wide. The roof height varies from approx. 4 m to 10.5 m. The internal room volume is approximately 25000 $m^3$, and was designed for 364 dairy cows (i.e., approx. 70 $m^3$ per animal). The barn has an open ridge slot, partly closed gable walls and open long sidewalls protected by nets and adjustable curtains (cf. Fig. 1). It represents a typical building design for moderate climate used, for example, in Northern Germany, the Netherlands or the Northern USA ((Mendes et al., 2015; Wang et al., 2016; Hempel et al., 2016b; Kafle et al., 2018)).





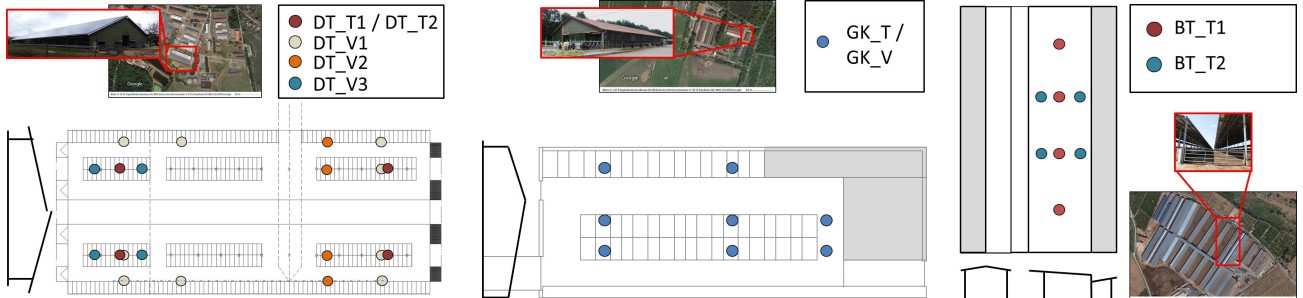

**Figure 1.** Outer view of the reference barn (source: ATB), aerial photo of the associated farm (source: Google Maps) and distribution of the sensors positions during different measurement campaigns for the three reference barns in Dummerstorf (left), Groß Kreutz (middle) and Bétera (right). In addition, the roof shapes are sketched. Details on the measurement campaigns can be found in Table. 1 (cf. IDs).

Temperature, relative humidity and air velocity were logged at various positions inside the barn in approximately 3 m height during measurement campaigns from summer 2015 until summer 2017 (cf. Table 1 and Fig. 1).

### 2.1.2 Reference barn Groß Kreutz - Central European continental region

This naturally ventilated dairy building is located in Eastern Germany (Teaching and Research Institute for Animal Breeding
and Animal Husbandry Groß Kreutz, Brandenburg, 32 m above sea level, cf. (Hempel et al., 2018)). It is approx. 39 m long and 18 m wide. The height of the roof varies from approx. 3.5 m to 6 m. The internal room volume is approximately 4500 m$^3$, designed for 50 dairy cows (i.e., ca. 90 m$^3$ per animal). The barn has a closed roof and partly closed gable walls (cf. Fig. 1). One long sidewall is open up to about 1.5 m height, the opposite is open up to the roof.

Temperature, relative humidity and air velocity were logged continuously at 8 positions inside the barn in approximately 3 m
height from summer 2015 until summer 2017 (cf. Table 1 and Fig. 1). In addition, at three of the sensor positions in the center of the barn, temperature and relative humidity was measured in 4 other heights between approximately 4 m and 6 m.

### 2.1.3 Reference barn Bétera - Western Mediterranean region

The commercial naturally ventilated building is located in Eastern Spain (More Holstein S.L, Bétera, Valencia, 125 m above sea level). It is approx. 137 m long and 18 m wide with open walls (fences) and a broad roof opening (cf. Fig. 1). The roof
height varies from approx. 4.5 m to 6 m. The internal room volume is approximately 12700 m$^3$, and was designed for 192 dairy cows (i.e., approx. 66 m$^3$ per animal).

Temperature and relative humidity were logged at various positions inside the barn in approximately 3 m height during two measurement campaigns in summer 2016 and 2017 (cf. Table 1 and Fig. 1).



## 2.2 Outdoor climate data

Our outdoor climate data consists of station observations from local weather services. We employed the observation network of the German Weather Service (DWD) for both reference barns in Germany. The observations for Spain were taken from the National Climatic Data Center (NCDC) Archive of the National Oceanic and Atmospheric Administration (NOAA). In detail,

observations for Valencia were based on Meteorological Aerodrome Reports (METAR) of Valencia Airport.

For each barn we selected one meteorological station based on two constraints: 1) the station had to be close to the respective barn to assure that the station weather observation is representative for the weather near the barn and 2) the station should cover at least the same period as the indoor measurements to permit a reasonable indoor model calibration. Table 2 summarizes the stations chosen for each reference barn respectively focus region.

**Table 2.** Stations and sources for meteorological outdoor data

| Focus region | Reference barn | Station name | Station ID | Source | Distance to barn |
|---|---|---|---|---|---|
| Central European maritime | Dummerstorf | Rostock-Warnemünde | 04271 | DWD Network | 21.29 km |
| Central European continental | Groß Kreutz | Potsdam | 03987 | DWD Network | 20.31 km |
| Western Mediterranean | Bétera | Valencia | 08284 | NCDC/NOAA Archive | 20.16 km |

All stations reported data with a temporal resolution of at least one hour. Where necessary, the unprocessed observations were aggregated to hourly values. Missing measurements were filled using a hot deck imputation method (Ford, 1983) based on temporal analogs. If no analog existed and the gap was smaller than 5 hours, a linear interpolation between measurement dates was used. The hot deck imputation method comprised all meteorological stations within a 150 km radius around the station under consideration.

To assess the impact of the anticipated climate change we used an ensemble of regional model projections. Table 3 summarizes the regional and driving global models used in our analysis. The simulations were partially conducted within the ReKliEs-De (http://reklies.hlnug.de) project and conform to the CORDEX-EUR11 specifications defined in the framework of the Coordinated Regional Downscaling Experiment (CORDEX, http://cordex.org, Giorgi and Gutowski Jr (2015)). Figure 2 presents the full simulation domain defined within this framework together with our three focus regions containing the refer-

ence barns. The simulations were available on a daily timescale with a horizontal resolution of 0.11 degree (approx. 12.5 km). They covered the period from 1970 to 2098. For each focus region we averaged the time series of the 9 grid boxes surrounding the meteorological station (cf. fig:map). The RCMs were driven by 8 different global climate models (GCMs) in total taking into account three different greenhouse gas concentration scenarios for the period 2006 to 2100, i.e. representative concentration pathways (RCPs) as defined in the IPCC's fifth Assessment Report (AR5) (Pachauri et al., 2014). Here, we considered an

anticipated radiative forcing of 2.6 W/m$^2$ (RCP 2.6), 4.5 W/m$^2$ (RCP 4.5) and 8.5 W/m$^2$ (RCP 8.5) in 2100. For the years 1970 to 2005 the RCMs used the observed greenhouse gas concentrations as boundary condition.

Before applying the simulations to our indoor climate model we adjusted the biases. For temperature and relative humidity we used the ISIMIP-FastTrack bias adjustment method using the station observations as reference (Hempel et al., 2013). The





**Table 3.** Modeling matrix of driving global models - rows - and the respective regional models - columns - used in this study. The numbers indicate the respective RCP scenarios covered by each GCM-RCM combination.

|  | SMHI-RCA4 | KNMI-RACMO22E | DMI-HIRHAM5 | CLM-CCLM4-8-17 | GERICS-REMO2015 | MPI-CSC-REMO2009 |
|---|---|---|---|---|---|---|
| HadGEM2-ES | 2.6 / 4.5 / 8.5 | 2.6 / 4.5 / 8.5 | - | 4.5 / 8.5 | 8.5 | - |
| EC-EARTH | 2.6 / 4.5 / 8.5 | 2.6 / 4.5 / 8.5 | 2.6 / 4.5 / 8.5 | 2.6 / 4.5 / 8.5 | 8.5 | - |
| MPI-ESM-LR | 2.6 / 4.5 / 8.5 | - | - | 2.6 / 4.5 / 8.5 | - | 2.6 / 4.5 / 8.5 |
| IPSL-CM5A-MR | 4.5 / 8.5 | - | - | - | - | - |
| CNRM-CM5 | 4.5 / 8.5 | - | - | 4.5 / 8.5 | 8.5 | - |
| NorESM1-M | - | - | 4.5 / 8.5 | - | - | - |
| CanESM2 | - | - | - | 8.5 | 8.5 | - |
| MIROC5 | - | - | - | 8.5 | 8.5 | - |

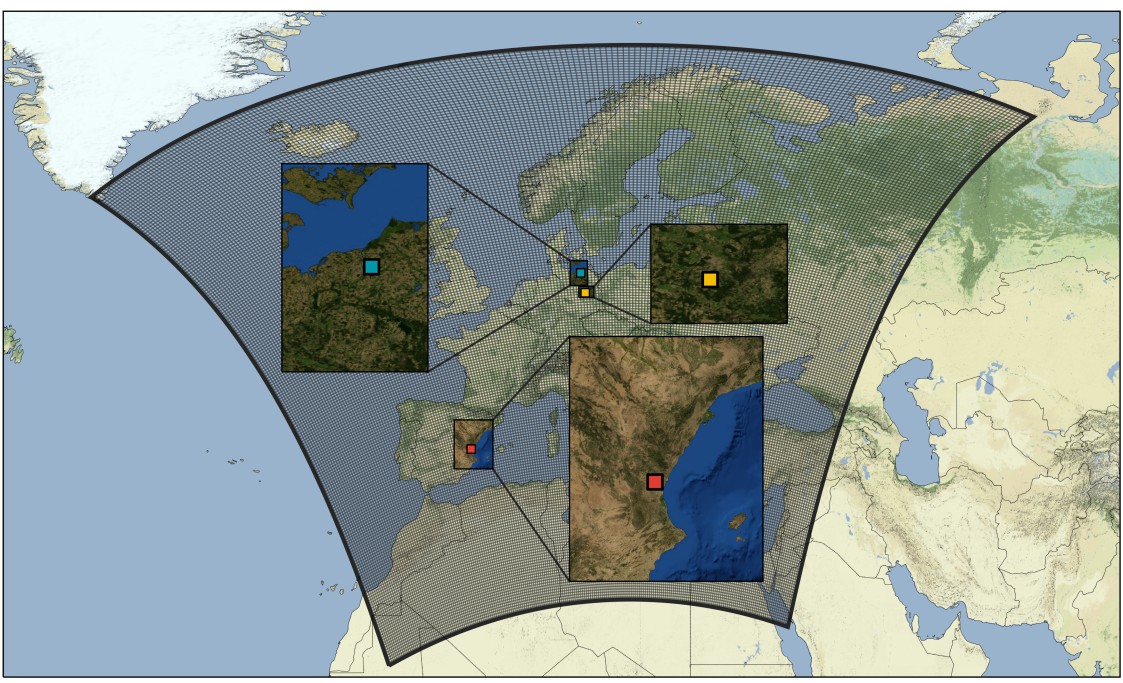

**Figure 2.** Map of the focus regions considered in our analysis. The surrounding lattice represents the RCM simulation domain (CORDEX-EUR11). The colored boxes mark the position of the selected meteorological stations of each focus region.

two horizontal wind components were adjusted together using a two-dimensional adjustment approach of Cannon (2018). Furthermore, we interpolated the daily values to hourly time steps using a regularized multivariate linear regression model. Each 24 hours of one day were mapped from daily values separately, i.e. our temporal downscaling does not account for inter-day dependencies.



## 2.3 Statistical indoor climate model

To analyze the impact of the anticipated climate change on the animal welfare and linked farm-economic and environmental factors, we need to determine the microclimate inside the barn from the outdoor weather conditions. Therefor we used a purely data-driven statistical approach, as it permits a fast and yet powerful simulation of the indoor microclimate conditions using the available observations. Depending on the statistical approach it implicitly implements the complex relationships between outdoor and indoor conditions depending on the building design, materials, orientation and outdoor environment. Compared to dynamical approaches it allows also an easy and automatized calibration for various climate conditions and barn layouts using individual configurations of observations (Gebremedhin and Wu, 2005; Wu et al., 2012; Fiedler et al., 2014).

We derived our statistical indoor climate model using outdoor weather conditions as predictors to estimate the microclimate inside each barn. We focused our modeling on those variables that have the most significant impact on animals' climate stress, namely temperature, relative humidity and wind (Mader et al., 2006) inside the barn. Solar radiation was neglected for our model as we assumed sufficient roof insulation and only minor radiation entries via the openings. In order to reduce the degree of complexity the indoor model uses only hourly and spatial averaged values of the whole barn (cf. discussion of uncertainty in calibration data in Sect. 3.3.4).

As our reference barns were naturally ventilated, the outdoor weather conditions significantly influence the indoor conditions. However the relationship is complex and based on non-linear physical processes. To simulate this relationship we tested several different statistical machine learning approaches. Due to the unique outdoor conditions in each region and the specific layout and building materials for each barn we trained a separate model for each barn. We tested the artificial neural network (ANN) approach, linear regression models with and without regularization, random forests regression and support vector regression models, all with different hyperparameter settings. The models were trained to predict at any hour the temperature, relative humidity and the two horizontal wind components inside the barn (predictands) based on outdoor temperature, relative humidity, zonal and meridional wind, sea level pressure and global radiation (predictors). For the Mediterranean region the model was set up with temperature and relative humidity as predictors and predictands only, due to the reduced observation data set for outdoor (Valencia) and indoor (Bétera) measurements. Each hour of the indoor predictands was modeled separately. Hence our model did not account for a memory effect of the indoor variables directly. However indirectly we considered a memory effect and lagged responses by using the values of the outdoor predictors two hours before and two hours after the predicted time step. Overall our feature space consists of 30 dimensions (6 predictors times 5 hours) for Dummerstorf and Groß Kreutz and 10 dimensions (2 predictors time 5 hours) for Bétera.

According to our analysis, the model type with the best performance was the ANN (Gurney, 1997; Heaton, 2015). To limit the complexity of the ANN we choose a simple dense network design. We tested different hyperparameter configurations of the ANN using up to three hidden layers with varying numbers of nodes in each layer and different activation functions. To train our ANN we used backpropagation (Werbos, 1974) with a mean squared error loss function. We prevented the network from overfitting by using a dropout regularization (Srivastava et al., 2014) and 8-fold cross validation. Table 4 summarizes the best performing ANN configuration for each reference barn along with the respective cross validation $R^2$ as performance score. For



| Reference barn | Layout | Activation | Predictor | Predictand | Total $R^2$ |
|---|---|---|---|---|---|
| Dummerstorf | (78, 54) | ReLU | T, H, W, P, R | T, H, W | 0.74 |
| Groß Kreutz | (90, 74) | ReLU | T, H, W, P, R | T, H, W | 0.56 |
| Bétera | (50) | ReLU | T, H | T, H | 0.85 |

**Table 4.** Best performing ANN configurations after a grid search hyperparameter optimization. We optimized the ANN for each reference barn separately. The layout configurations refer to the number of nodes in each hidden layer. The predictor and predictand abbreviations are defined by: T - temperature, H - relative humidity, W - zonal and meridional wind, P - sea level pressure, R - global radiation.

all reference barns we found a rectified linear unit (ReLU) activation function in each node as the optimal choice. The output layer uses a linear activation function as it maps to the real valued predictands. The final network layout for Dummerstorf and Groß Kreutz is more complex compared to Bétera due to the lower number of predictors and predictands for the latter barn. The different predictands are also reflected in lower $R^2$ scores for the German barns with $0.74$ and $0.56$ for Dummerstorf and
Groß Kreutz compared to $0.85$ for Bétera. The low performance of the wind components inside the barn resulted in the lower total $R^2$ score.

## 2.4 Statistical evaluation

As shown in table 3, our RCM ensemble was imbalanced towards CLM-CCLM4-8-17, GERICS-REMO2015 and SMHI-RCA4 regional models, with six and five driving GCMs respectively compared to only two driving models for KNMI-RACMO22E
and DMI-HIRHAM5 and one for MPI-CSC-REMO2009. This imbalance would propagate into our uncertainty estimation if we naively assume that each simulation, i.e. each GCM-RCM combination, is equally weighted in the ensemble. However our assumption is that each single RCM should be equally weighted. We think of the different GCM simulations driving the same RCM as an additional artificial variability of that RCM. Hence we consider 3 different sources of uncertainty in our statistical evaluation of the full model ensemble:

– Temporal uncertainty. Estimated from the year-to-year variability of the time series of each single simulation.

– GCM uncertainty. Estimated from different GCMs driving the same RCM.

– RCM uncertainty. Estimated from the different RCM simulations.

All these sources are coupled and the underlying probability distribution is not necessarily Gaussian. To avoid the usage of a complex statistical model we adopted a simple bootstrap method for the statistical evaluation of the full ensemble (Efron,
1979; Efron and Tibshirani, 1986).

We draw 10000 random samples of the GCM-RCM matrix keeping the same structure (i.e. the same number of total RCMs, the same number of GCMs per RCM). This is done in three steps. In the first step we draw 6 RCMs out of the 6 available. In the second step we draw randomly from the available GCMs of the respective RCM. Here we draw the same number of times as GCMs are available for that RCM (five for SMHI-RCA4, two for KNMI-RACMO22E, two for DMI-HIRHAM5, six for CLM-





CCLM4-8-17, five for GERICS-REMO20015 and one for MPI-CSC-REMO2009). In the last step we draw randomly from the available reference years (30 out of 1971-2000) for each RCM-GCM combination. This way we end up with a bootstrap sample with the same structure and magnitude as the original modeling matrix. For each sample we calculate first the time average. Then we average over all GCM simulations of one RCM. We end up with 6 averaged RCMs for each sample. This

way we can estimate the spread of the ensemble considering three uncertainty sources and their propagation.

## 2.5  Empirical heat stress models

Heat stress conditions for dairy cattle are expected to occur much more frequently in our focus regions than cold stress conditions. Hence, we focused our assessment on heat stress. To quantify the effect of the indoor mircoclimate on the animal perceived stress we considered two different heat stress indices. These empirical models link air temperature with additional

state variables of the indoor air in order to evaluate the anticipated individual comfort or discomfort under hot environmental conditions (e.g., hot and humid air appears particularly warm). The two selected indices differ according to the number of state variables considered. Heat stress was evaluated in term of thresholds (cf. Table 5).

First, we considered the temperature humidity index (THI, cf. Eq. 1) originally published in this form by the United States National Weather Service (NRC, 1971). Since the early 1990's the index was frequently used to evaluate heat stress in cattle

using the following definition:

$$THI = (1.8 \cdot T + 32) - ((0.55 - 0.0055 \cdot H) \cdot (1.8 \cdot T - 26)) \tag{1}$$

with air temperature (T) in °C and relative humidity (H) in %. The THI incorporates dry bulb temperature and relative humidity, but it does not take into account wind speed or solar radiation (Dikmen and Hansen, 2009; Lees et al., 2018; Wang et al., 2018b).

Second, we considered the equivalent temperature index for cattle (ETIC, cf. Eq. 2) which was developed within the OptiBarn

project (Wang et al., 2018b). We used the following ETIC definition:

$$ETIC = T - 0.0038 \cdot T \cdot (100 - H) - 0.1173 \cdot |\boldsymbol{v}|^{0.707} \cdot (39.20 - T) + 1.86 \cdot 10^{-4} \cdot T \cdot Q \tag{2}$$

with air temperature T in °C, relative humidity H in %, air velocity $\boldsymbol{v}$ in $m\,s^{-1}$ and solar radiation Q in $W\,m^{-2}$.

For the evaluation of the heat stress risk, we defined the number of heat stress events (HSE) as amount of hours of at least moderate heat stress, i.e. with indoor THI $\geq$ 72 respectively ETIC $\geq$ 20 (cf. Table 5). In addition, we considered the duration

of heat stress events (HSED) as the length of periods of consecutive hours of at least moderate heat stress. In the analysis of the heat stress risk we considered changes compared to a reference period 1971-2000.

## 2.6  Impact assessment

Heat stress is known to affect farm-economics (e.g. milk yield and quality) and environment (emissions) as well as animal physiology and behavior resulting in impacts on animal welfare and health. We used physiological and behavioral data collected

in the OptiBarn project as well as contemporary literature for the impact assessment.



**Table 5.** Categories of heat stress levels used for our assessment. For THI we used the thresholds defined by Collier based on milk yield losses (Collier et al., 2012). For ETIC we used the thresholds defined by Wang based on a linear regression between THI and ETIC without wind (i.e., $v = 0 \ m \, s^{-1}$) and solar radiation (i.e., $Q = 0 \ W \, m^{-2}$) (Wang et al., 2018a)

|      | mild stress | moderate stress | severe stress | emergency |
|------|-------------|-----------------|---------------|-----------|
| THI  | $68 \leq \text{THI} < 72$ | $72 \leq \text{THI} < 80$ | $80 \leq \text{THI} < 90$ | $\geq 90$ |
| ETIC | $18 \leq \text{ETIC} < 20$ | $20 \leq \text{ETIC} < 25$ | $25 \leq \text{ETIC} < 32$ | $\text{ETIC} \geq 32$ |

Since the underlying assumptions and compiled data in the quantification of the heat stress impacts introduce considerable additional uncertainty, we will only concentrate on the magnitude of impacts. We focused our assessment on the RCP 8.5 scenario to estimate impacts under the strongest anticipated climate change. Furthermore, we neglected the range of uncertainty in the model projections and potential adaptation measures (related, for example, to housing, feeding or breeding). Moreover,

the physiological adaptation (due to a general temperature increase and prolonged heat load duration), and the effect of heat load aggregation over the day, i.e. the number of heat stress hours where the THI was above the onset of mild heat stress (i.e., THI $\geq 68$, cf. Table 5) was not taken into consideration (St-Pierre et al., 2003). The daily maximum THI was not explicitly considered and the further aggravation of impacts for THI>72 was neglected.

For the estimation of impacts, where the increase or decline rates were given per THI unit in literature, we used a factor 4

to scale the rates as in our results we considered the increase of at least moderate heat stress events (i.e. THI$\geq$72 relative to a heat stress threshold of 68, namely at least 4 THI units).

As many impacts were related to daily THI values in literature, we need to estimate the number of heat stress days (HSD) based on the hourly THI values. This, however, requires the introduction of additional constrains (e.g., what's the minimal number of heat stress hours to make a heat stress day). To simplify the accumulation of hourly heat stress events to daily mean

heat stress we assume that only one period of consecutive heat stress hours occurs per day. In consequence, we approximate the number of heat stress days by dividing the average projected number of heat stress events by its duration (i.e., HSE/HSED).

Note that regional differences in the impacts might be underrepresented in the overall estimation as there was only data for three barns available.

## 3 Results and Discussion

In the following section, we present projections of the indoor climate in the reference barns in our three focus regions as well as the deduced heat stress risk for the housed dairy cattle. For the sake of simplicity, we focus on the results of individual barns to highlight the seasonal characteristics as well as differences between the three RCP scenarios and the two stress indices. We discuss levels of uncertainty in our projections of heat stress risk. Moreover, we outline resulting regional impacts and potential adaptation measures.





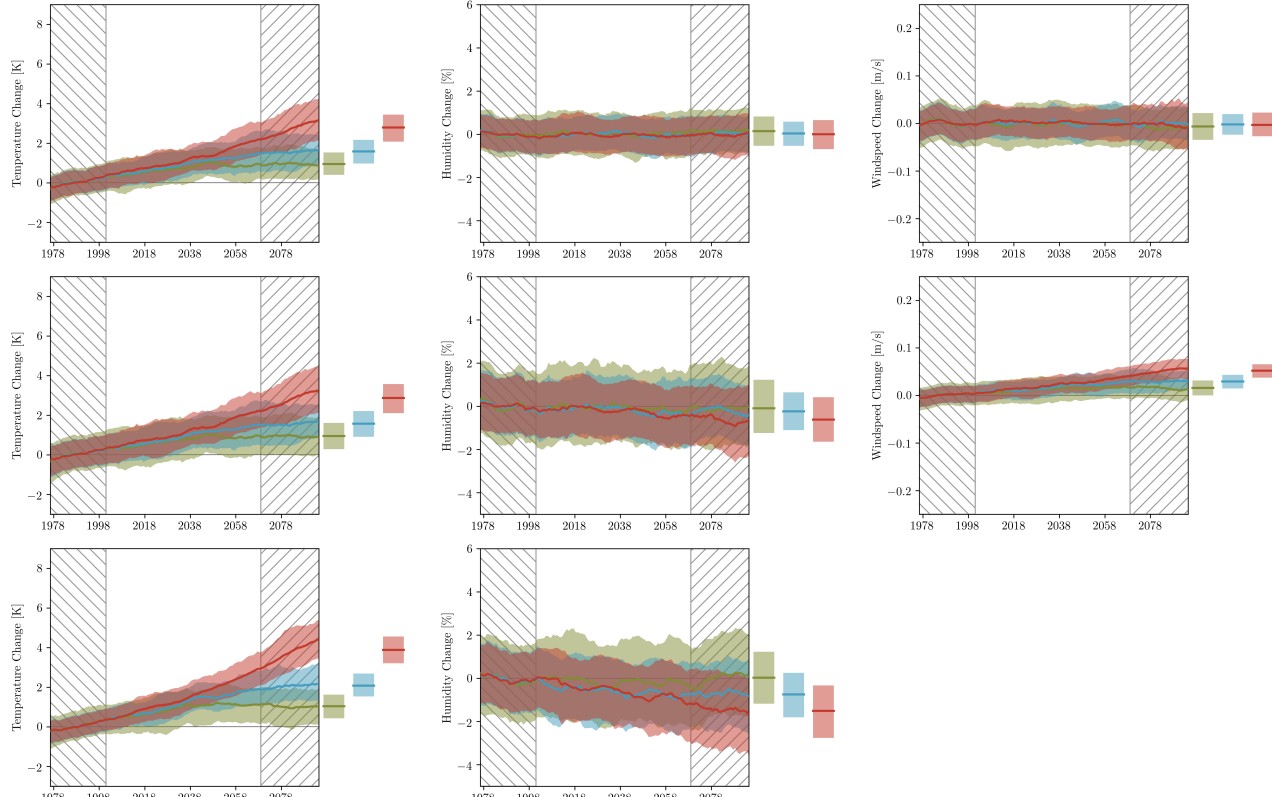

**Figure 3.** Projected change in indoor temperature, humidity and wind in the three focus regions of this study under RCP 2.6 (green), 4.5 (blue) and 8.5 (red). Regions: Central European maritime region with reference weather station Rostock-Warnemünde (top), Central European continental region with reference weather station Potsdam (middle) and Western Mediterranean region with reference weather station Valencia (bottom).

### 3.1 Indoor climate changes

The projected climate change and thus the anticipated indoor climate differed depending on the greenhouse gas concentration scenario and the region under consideration. Overall the temperature is expected to increase in all three focus regions until the end of the century between $1\,°C$ and $5\,°C$. The statistical model simulations showed a slightly higher increase of indoor

5 temperatures for the barn in the Western Mediterranean compared to those in the Central European focus regions (cf. Fig. 3).

The anticipated relative humidity remained approximately constant in the barn in the Central European maritime region and decreased towards the South (with a stronger decrease for the barn in the Mediterranean region than for the barn in the Central European continental region). All changes were well below $5\,\%$ and in most cases not statistically significant. However, the decrease in the Mediterranean region is inline with the anticipated temperature increase and precipitation decrease.

10 For the reference barn Groß Kreutz (Central European continental region) the climate model ensemble projected a slight increase in the near-surface wind under all three RCP scenarios. The RCP 8.5 showed the largest increase in the average indoor





wind speed with a value of up to $0.05\,\mathrm{m\,s^{-1}}$ over the century (cf. Fig. 3). This means a decrease by more than 10% taking into account that the typical average wind speed in the barn today is around $0.4\,\mathrm{m\,s^{-1}}$. The change was particularly pronounced in the summer months with up to approximately $0.15\,\mathrm{m\,s^{-1}}$ during June, July and August compared to almost $0\,\mathrm{m\,s^{-1}}$ in December, January, February (results not shown).

In contrast, in the reference barn Dummerstorf (Central European maritime region) no significant annual trend in the average wind speed was found (cf. Fig. 3). Largest changes were projected in autumn (up to $-0.1\,\mathrm{m\,s^{-1}}$) and winter (up to $0.15\,\mathrm{m\,s^{-1}}$) under RCP 8.5, but model simulations were discordant with regard to the trends (results not shown).

    Wind projections for the reference barn Bétera (Mediterranean region) were not available due to a lack of sufficiently long wind measurements in the barn.

## 10  3.2   Heat stress risk

The number and duration of heat stress events was derived from the indoor climate projections. The risk of moderate heat stress showed an overall increasing trend.

### 3.2.1   Risk under different RCPs

In order to assess the effect of different atmospheric greenhouse gas concentrations on the heat stress risk, we considered the
example of the reference barn Groß Kreutz (Germany, Central European continental region) as for this location we could make use of the most comprehensive and homogeneous data set. The heat stress risk under RCP 2.6, RCP 4.5 and RCP 8.5 was evaluated using the number and duration of anticipated heat stress events based on the projected indoor THI as described in Sect. 2.5 (cf. Fig. 4).

    Until mid of the 21st century ($\approx$ 2040) there is no significant difference between the projections under different RCP scenar-
ios. The average duration of the stress events is expected to increase up to approximately 1 h in all scenarios, while the number of events is expected to increase by up to approximately 150 (i.e., up to approximately 2% of all hours of a year, corresponding to 6% of all hours of a summer, will be classified as at least moderate stress event in addition to the current situation).

    For the second half of the 21st century, the average duration of heat stress events is expected to stay at the mid-century level under RCP 2.6 and RCP 4.5. For the extreme scenario RCP 8.5, the increase continues up to approximately 3 h. The number
of heat stress events is projected to stay approximately at the mid-century level under RCP 2.6 and increases only slightly during the second half of the century under RCP 4.5 (approximately 150 additional heat stress events for RCP 2.6 and 200 for RCP 4.5). Under RCP 8.5, however, the number of additional heat stress events increases up to approximately 600, i.e. nearly 7% of all hours of a year will be additionally classified as heat stress events, most pronounced in the summer season. Hence, approximately 27% of all summer hours, i.e. more than every fourth hour, will be characterized by at least moderate heat stress
conditions in addition by the end of the century.

    Despite the described relations between the ensemble averages for each RCP scenario there was a range of uncertainty of approximately $\pm 1\,h$ regarding the duration and $\pm 200$ regarding the number.



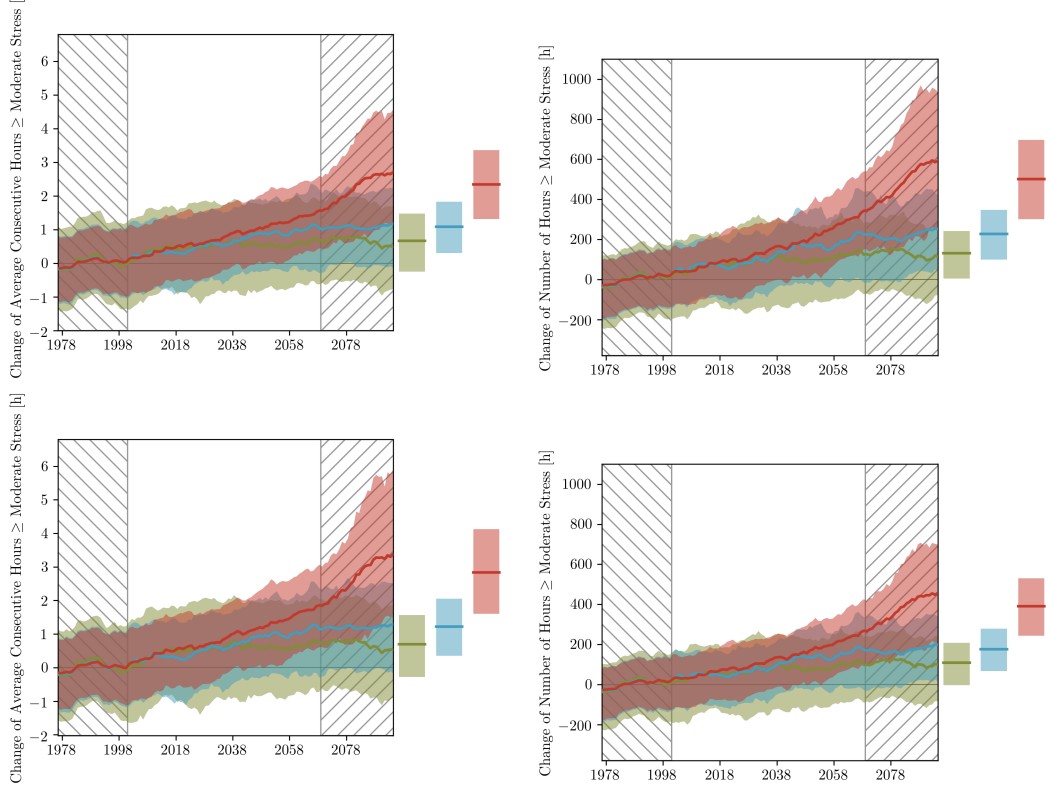

**Figure 4.** Projected change in heat stress events in the reference barn in Central Europe (annual changes in upper panels and summer changes in lower panels). Modifications in average duration (average consecutive hours) and number (number of consecutive hours) of moderate stress events under RCP 2.6 (green), 4.5 (blue) and 8.5 (red) are plotted based on the evolution of the temperature humidity index (THI) with a threshold of 72.

### 3.2.2 Regional differences

In order to evaluate the regional differences of climate change impacts on dairy farming, we considered the example of two reference barns in maritime regions, one in Central Europe and one in the Mediterranean region (cf. Fig. 5). As in Sect. 3.2.1 the heat stress risk under RCP 2.6, RCP 4.5 and RCP 8.5 was evaluated in terms of number and duration of heat stress events as described in Sect. 2.5.

Although the annual temperature increase does not differ a lot among the regions (Fig. 3) the implications in terms of critical THI values were rather different for the two regions. By mid of the 21st century we found an increase of the duration of heat stress events of approximately 1.5 h for Central Europe and 2.5 h for the Mediterranean region under all RCPs. For RCP 2.6 this is also true until the end of the century. Under RCP 4.5 in the second half of the century an increase of 2 h for Central Europe versus 5 h for the Mediterranean region was projected. For RCP 8.5 the deviation between the regions is even larger





**Figure 5.** Projected change in heat stress events in the reference barns in maritime regions in Central Europe (2 top rows) versus Mediterranean region (2 bottom rows). Modifications in annual (first row) and summer (second row) average of duration (average consecutive hours) and number (number of consecutive hours) of moderate stress events under RCP 2.6 (green), 4.5 (blue) and 8.5 (red) are plotted based on the temperature humidity index (THI) with a threshold of 72.





with an increase of 4 h in Central Europe versus 17 h in the Mediterranean region. The latter implies that there will be barely any recovery phases for the cows in the reference barn in the Mediterranean region.

The change in the number of the heat stress events was even more divers among the regions and RCPs. While for the barn in Central Europe up to approximately 200 additional hours of heat stress are expected (i.e., even less than for the barn in

the Central European continental region described in the previous subsection), for the Mediterranean region the increase was much stronger: Under RCP 2.6 approximately 300 additional heat stress events (i.e., 4% of all hours of a year), under RCP 4.5 approximately 800 additional heat stress events (i.e., nearly 9% of all hours of a year) and under RCP 8.5 approximately 1800 additional heat stress events (i.e., nearly 21% of all hours of a year) were projected. In addition, in Central Europe most of the additional heat stress events occurred in the summer season. In the Mediterranean region, only approximately half of the

increase was anticipated for summer, while there was a significant increase in the number of heat stress events projected for spring and autumn.

Again, we focused the comparison on the ensemble averages, while there was a range of uncertainty of $\pm 200$ regarding the number and approximately $\pm 2$ $h$ (Central Europe) respectively $\pm 5$ $h$ (Mediterranean) regarding the duration. Towards the end of the century, the ranges tend to further increase, particularly for the RCP 8.5 scenario.

### 3.2.3   The effect of air movement

We investigated the impact of wind as additional environmental parameter to evaluate the heat stress risk using the example of the reference barn Groß Kreutz (Germany, Central European continental region) and the heat stress index ETIC.

The change in duration and number of heat stress events was investigated using ETIC with and without wind, neglecting radiation effects in both cases (cf. Fig. 6). We found a general tendency towards an increase of the duration and number of

heat stress events by the end of the century which was in the same order of magnitude as predicted using the THI (cf. Fig. 6 compared to Fig. 4).

Without the wind effect (i.e., ETIC with $v = 0$ $m\,s^{-1}$ at all time points, cf. Fig. 6 upper panels) the projected increase of the duration of heat stress was under RCP 8.5 on average 4 h (ETIC) compared to 3 h (THI). Taking the range of the model ensemble into consideration, it was even 6 h (ETIC) versus 4.5 h (THI). Similar deviations were observed for the estimated

increase in the number of heat stress events. Here, ETIC under RCP 8.5 resulted in on average 800 more heat stress events (up to 1200 considering the uncertainty of the model ensemble) while THI resulted in on average 600 more heat stress events (up to 900 considering the uncertainty of the model ensemble).

Taking into account the wind in the ETIC calculation (cf. Fig. 6 lower panels), the heat stress risk decreased as the wind increased (particularly in summer), which had a cooling effect. While ETIC under RCP 8.5 projected on average approximately

800 additional heat stress events without wind effect, on average approximately 600 additional heat stress events were projected with wind effect. This is a relative reduction of the number of heat stress events of approximately 25% due to changes in the wind regime. However, the projected heat stress risk considering ETIC with the wind effect was almost the same as that projected with the THI which does not include the wind effect at all.





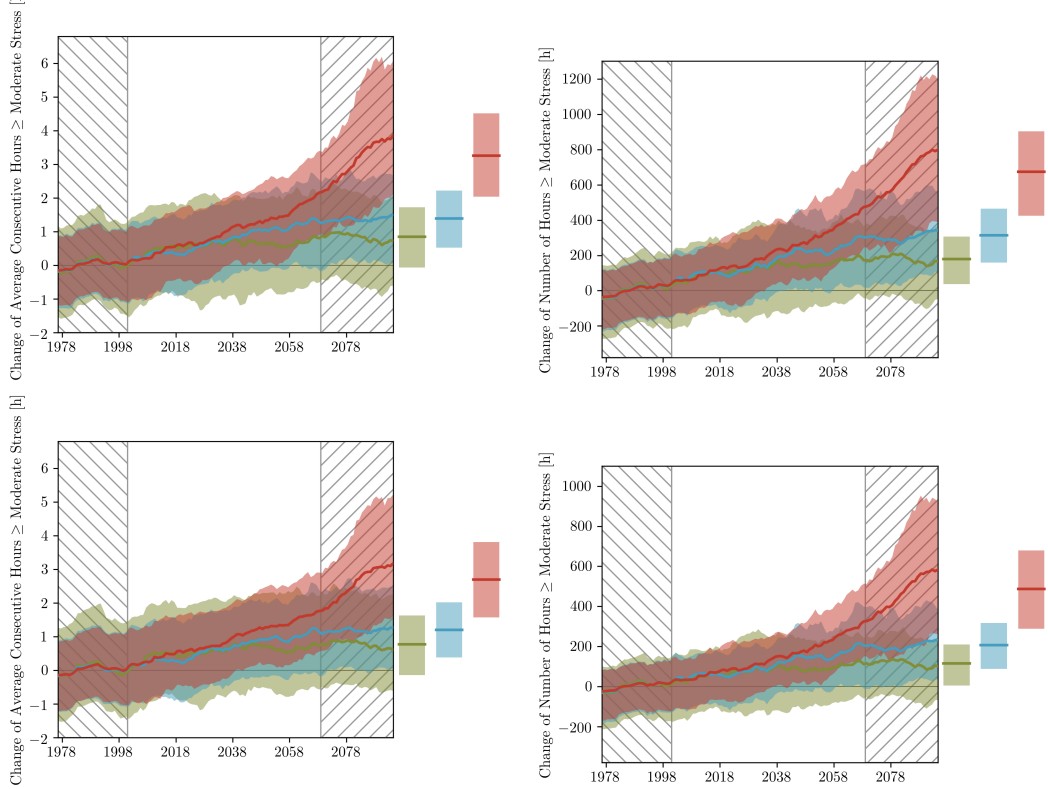

**Figure 6.** Projected change in heat stress events in the reference barn in Central Europe without (top) and with (bottom) consideration of near-surface wind. Modifications in annual average of duration (average consecutive hours) and number (number of consecutive hours) of at least moderate stress events under RCP 2.6 (green), 4.5 (blue) and 8.5 (red) are plotted based on the Equivalent Temperature Index for Cattle (ETIC) with a threshold of 20.

## 3.3 Model uncertainty

While our results showed a general tendency of increasing heat stress risk, there was a large uncertainty on the magnitude of the anticipated increase. The overall model uncertainty involves: (1) the climate models used to drive the indoor climate model, (2) the considered greenhouse gas concentration scenarios, (3) the selected index and threshold that was used to define heat
5 stress events, and (4) the accuracy of measurements used to calibrate the indoor climate model. In the following, we discuss these various sources of model uncertainty in our study qualitatively.

### 3.3.1 Climate model ensembles

The span of anticipated climate signals was well represented in our ensemble with GCMs projecting strong respectively weak changes in the global mean temperature and precipitation (Warszawski et al., 2014). The RCM-GCM combinations capture





the edges of the intermodel variability in the CORDEX-EUR ensemble in terms of biases in temperature and precipitation (cf. (Dosio, 2016)).

However, only a limited number of runs per RCP was available for our study, namely 11 for RCP 2.6, 12 for RCP 4.5 and 21 for RCP 8.5. Increasing the number of simulations might increase our estimated uncertainty ranges. In order to account for
the imbalanced sampling among the possible climate signals (cf. sparse modeling matrix in Table 3), we used a bootstrapping algorithm for the statistical analysis to evaluate the heat stress risk. Our analysis shows a similar range of the anticipated changes in the number of heat stress events for all three barns as well as all RCPs and time slices. The uncertainty in the anticipated duration of heat stress events was also similar among the RCPs and time slices.

The uncertainty in the duration, however, varied regionally. In the Mediterranean region, where the largest impacts are
expected, it was particularly high, yielding additional challenges for adaption. These regional differences in the uncertainty might be, however, to some degree also caused by the fact that the calibration data sets for the indoor-THI-model differed with regard to comprehensiveness and homogeneity.

### 3.3.2 RCP scenarios

Earlier other authors used one climate model, one emission scenario and the outdoor THI to perform a similar study for Spain
(Segnalini et al., 2013). Our study on the other hand was based on a model ensemble and used an indoor THI together with multiple recent emission scenarios to cover the full range of future socio-economic pathways.

Segnalini et al. (2013) derived a slight increase of the heat stress risk mainly in summer . Our results based on the indoor THI under RCP 2.6 were well inline with this former study. However, based on the more recent climate projections, a larger ensemble of climate models and incorporating the effect of housing on the THI, our results also indicated that the former
projections can be understood as some kind of best case scenario. The projected indoor temperature change by the end of the 21st century deviated in our study among the RCPs by approximately $2\,°C$ in the barns in Central Europe and $4\,°C$ in the barn in the Mediterranean region (cf. Fig. 3). At the same time, the projected change of annual humidity deviated by less then 3% relative humidity. Thus, the THI increase is likely to be higher than previously assumed.

Moreover, the projected change in temperature, humidity and wind resulted in an increase of the duration of heat stress
events which was approximately $3\,h$ (Central European regions) respectively $17\,h$ (Mediterranean region) less under RCP 2.6 than under RCP 8.5. In addition, approximately 5% heat stress hours less per year (relative to the total number of hours in a year) were projected under RCP 2.6 compared to RCP 8.5 for the Central European regions by the end of the century. In the Mediterranean region the deviation between RCP 2.6 compared to RCP 8.5 was approximately 17%. This implies that with the continuation of the global warming the regional differences will be amplified with higher greenhouse gas concentrations.

The RCP scenarios considered in our study were associated with different socio-economic, technological and political developments that yield different atmospheric greenhouse gas concentrations. Feedback loops induced by changing duration and number of heat stress events were not included. However, it is known that the net emissions from cattle husbandry (dependent on the storage, treatment and application of manure and slurry as well as on the production level and feed of the ruminants) are influenced by environmental parameter such as temperature and humidity (Monteny et al., 2001). Increased heat stress




may yield higher ammonia and methane emissions affecting aerosol formation and amplifying the increase in greenhouse gas concentration as discussed in Sect. 3.4.2. Such effects are, however, underexplored so far and should be addressed in future studies to further develop representative greenhouse gas and ammonia concentration scenarios for climate impact assessments.

### 3.3.3 Stress index

A lot of cattle-related thermal indices have been proposed in literature, two of which we considered in our study (Bianca, 1962; Mader et al., 2006; Gaughan et al., 2008; Mader et al., 2010; Da Silva et al., 2015; Lees et al., 2018; Wang et al., 2018b). The genetic basis used to derive the indices relative to the dominating genotypic conditions in our study (including resilience types and future adaptation) represent additional incalculable factors. Our results (cf. Sect. 3.2.3) indicated that the uncertainty that was introduced by choosing one of the stress indices (THI or ETIC with respectively without wind) together with a particular

threshold was in the same order of magnitude as the wind effect. This leads to the assumption, that the effect of wind speed could be neglected in heat stress risk projections using the common indices.

It has, however, to be noted that these results referred only to an averaged wind speed in the barn under consideration, while the distribution of the air flow inside the building is very sensitive to changes in the inflow conditions (e.g. surrounding building or planting) as well as the building design (cf. Sec. 3.3.4) (Hempel et al., 2015b; Yi et al., 2018). The impact of including or

neglecting the wind may be different for other barns or even when considering heat stress levels in individual locations of the same barn.

The location of the animals inside the barn is also crucial with regards to shading as the lack of shade shifts the heat stress threshold towards lower ambient temperatures (e.g., body temperature increases earlier) (Berman, 2005; Kendall et al., 2006). This effect was neglected in our study as the cows were free to move inside the barns and could look for shade almost

all the time. The roof height and insulation were considered sufficient to avoid direct radiation effects. The validity of this approximation depends on the building design. Larger radiation effects can be expected in the reference barn in Bétera (Spain, Western Mediterranean region) which has a wide roof opening compared to the reference barn in Groß Kreutz (Germany, Central European continental region) which has no roof opening, but only roof-lights.

Moreover, the projected indoor temperature, humidity and wind were not linearly translated into heat stress events, as in

our study heat stress was evaluated in terms of critical thresholds. The choice of the threshold affects the risk assessment and depends on the considered indicator (e.g., milk yield, body temperature, respiration rate, rumination or lying time). The threshold we used for THI was based on milk yield depression and related to respiration rate in literature (Bohmanova et al., 2007; Collier et al., 2012). Although based on the same indicator, the thresholds for THI and ETIC differ slightly due to the imperfect correlation and rounding when ETIC thresholds were derived from THI thresholds with a linear transfer function

(Wang et al., 2018b).

In addition, the thresholds provided in literature only refer to some kind of prototypical cattle. The actual threshold is animal-individual and depends on the local environment to which the animals are adapted (Bohmanova et al., 2007). Referring to the thresholds defined by Collier (cf. Sec.2.5), for example, Israeli Holstein in summer in Israel are permanently under heat stress conditions, while producing a similar amount of milk or even more than Holstein Frisian in Germany, as breed, barn design



and cooling management are already adapted to the local climate conditions (Pinto et al., 2019). In consequence, due to the regional adaptation the threshold for moderate stress in our Central European focus regions may relate only to mild stress in our Mediterranean focus region.

Furthermore, under heat stress conditions cows tend to adapt their body posture together with the respiration rate (Pinto et al., 2017). As laying down decreases the body surface area of cows that is exposed to air by approximately 42 % the effect of wind differs depending on the body posture. In consequence, lying cows increase the respiration rate significantly more than cows in standing posture and cows with higher core body temperature tend to stand up (Allen et al., 2015; Hempel et al., 2016a). As body posture changes depending on the barn management (e.g., feeding or milking times) and in reaction to the ambient climate conditions may occur simultaneously additional uncertainty in the threshold selection is introduced.

Finally, milk yield, lactation phase and age influence directly the heat stress susceptibility. Because of the metabolized energy used for milk production, high-yielding cows have significantly more heat to dissipate than low-yielding cows and are thus more susceptible to heat stress (Kadzere et al., 2002; West, 2003; Spiers et al., 2004). Cows at the beginning of the lactation tend to produce more milk than cows during a late lactation period, yielding higher heat stress susceptibility. In addition, there are evidences that older (multiparous) cows are more affected by heat load than young (primiparous) ones (Bernabucci et al., 2014). As the herds in our reference barns consist of cows of different age and lactation phase and hence different milk yield, the projected heat stress risks will apply only for herd average.

With the breeding towards higher milk yields of the individual cows, as observed in the past, the thresholds for heat stress can be expected to further decrease on a herd level (Carabano et al., 2016). This will amplify the projected heat stress risks. However, further quantitative studies are required to evaluate to which degree the thresholds will decrease in the future in the focus regions.

### 3.3.4 Calibration data

Besides systematic errors related to the measurement device accuracy and long-term stability, the spatial and temporal variability of the outdoor and indoor measurements, that were used to derive and calibrate our indoor climate model, introduces uncertainty related to the sampling procedure.

For outdoor measurements (i.e., weather observations from meteorological observation stations) this relates mainly to the question if the weather conditions at the meteorological station are connected to those near the barn. Here the aspect of closeness is crucial, but also topographic / orographic considerations which affect particularly the dynamics of the humidity time series (e.g., the autocorrelation function decreases faster for Rostock-Warnemünde than for Dummerstorf potentially because it is closer to the sea (Hempel et al., 2018)). In order to minimize the impact of such complex effects, we applied the artificial neural network approach and considered a time lag of $\pm 2$ h (cf. Sect. 2.3).

For indoor measurements the topic of data variability relates to the question if point measurements are representative for the barn average. All measurement devices were mounted in such a way that they were as much as possible exposed to free inflow in order to minimize systematic errors induced by the construction material of the barns (e.g. wind shading or conduction). Moreover, to reduce the spatial and temporal variability in the relevant variables, we focused our modeling on hourly averaged



values. Direct air movement and turbulent diffusion, which are mainly responsible for the high spatial variability, can not be resolved on hourly timescale, which justifies also our choice of focusing on spatial averages over the whole barn.

The calibration data used for our statistical indoor models differed in homogeneity and extent between the individual barns. The most comprehensive data set was the one for the reference barn Gross Kreutz where wind, temperature and humidity were
measured in about 3 m height approximately every 10 m. In contrast, in the reference barns Dummerstorf and Betera the spatial resolution of the measurements and the spatial coverage of individual barn locations changed over the measurement periods and differed between the variables. This introduces additional uncertainty in the spatial averages. In addition, three vertical indoor profiles for temperature and relative humidity were incorporated in the spatial averaging in Gross Kreutz which were not available for the other barns. The observed vertical gradients in temperature and relative humidity are, however, much
smaller than the horizontal variability. Thus, the effect of this additional data can be neglected for the total barn average.

Preceding studies indicated that there is a high degree of turbulence yielding to significant spatial inhomogeneities in the velocity field inside the barn (Fiedler et al., 2014; Yi et al., 2018). Depending on the opening sizes of the inlets eddies of different sizes propagate towards the outlets (Hempel et al., 2015a; Yi et al., 2018). The spatial spreading of the wind speed $\tilde{v}$ increases typically with the average wind speed $\bar{v}$ (i.e., $\tilde{v} \approx 6.96 \cdot \bar{v}$ for the barns under consideration). This means the spatial
variability of the wind depends on the actual inflow speed. Considering the median of the distribution of the observed hourly averaged wind speed values, the typical spatial wind speed variance is approximately $2.4\,\mathrm{m\,s^{-1}}$. The anticipated changes in the near-surface wind, which governs the inflow, are however regionally very divers (Kjellström et al., 2018).

Considering the threshold ETIC = 20 (as used in our study, a decrease of the wind speed from 2.4 m/s to 0 m/s results in a increase of the ETIC value between approximately 1.5 (for very low relative humidity) and 4 for very high humidity. This
means under arid climate cows in some locations of the barn, that are particularly exposed to the wind, will only exceed the threshold for mild stress, while others in calm locations suffer from already moderate stress. Under humid climate, the effect is even stronger as cows in locations that are particularly exposed to the wind may not even be under heat stress at all (ETIC with wind approximately 16 instead of 20), while others are suffering from already moderate heat stress.

The inhomogeneous distribution of heat and humidity sources related to farm management and the turbulent inflow associ-
ated with the meteorological boundary conditions yield also high spatial and temporal variability of air temperature (approximately $\pm 2\,^{\circ}\mathrm{C}$) and relative humidity (up to $\pm 20\,\%$ relative humidity) (Herbut et al., 2015; Hempel et al., 2018). For THI values close to the threshold 72 each of those uncertainty values correspond approximately to $\pm 2$ THI units. However, this estimated spatial variability refers to measurements with a temporal resolution of 5 to 10 minutes.

However, it has to be noted that the spatial variability of the THI inside the barn can be up to $\pm 4$ THI units from the projected
average value (considering the spatial variation of temperature and humidity as independent of each other). This is almost a difference of one THI class, which implies that events classified as moderate based on the spatial average might correspond to already severe heat stress in some locations and only mild stress in other parts of the building.



## 3.4 Projected impacts and adaption options

The anticipated increase of the number and duration of heat stress events will have significant impacts and implies a strong need for adaptation measures for the European dairy husbandry system due to economic, environmental and ethic (animal welfare) aspects.

### 5   3.4.1   Economic impact

Heat stress affects the reproductive performance of cows and decreases fertility (De Rensis et al., 2015; Schüller, 2015). In addition, elevated temperatures may increase disease pressure and lead on to more health treatment (cf. Sect. 3.4.3). In particular, the occurrence of heat stress events can be translated into losses in milk yield and quality where the decline of milk yield begins directly after being exposed to uncomfortable environmental conditions (Rushen et al., 2001). In literature

it is often assumed that milk yield stays almost constant until a certain threshold and then linearly declines with increasing degree of THI (Ravagnolo and Misztal, 2000; Bohmanova et al., 2007). The decline rate per cow and day per THI unit with the onset of heat stress has been estimated for Holstein dairy cattle (the breed in our reference barns) to be, for example, between $0.3\,\mathrm{kg}$ and $0.39\,\mathrm{kg}$ for cows with $28\,\mathrm{kg}$ daily milk yield in the US and around $0.41\,\mathrm{kg}$ for cows with $20\,\mathrm{kg}$ milk yield in the Mediterranean region (Bouraoui et al., 2002; Bohmanova et al., 2007). Higher-yielding cows are known to be more susceptible

to heat stress (Kadzere et al., 2002; Nardone et al., 2010; West, 2003). It has been shown that at the same THI threshold the most productive cows (yield average $42.5\,\mathrm{kg\,d^{-1}}$) of the same Holstein breed lost $0.174\,\mathrm{kg\,d^{-1}}$ per unit of THI more than the average cows ($31.5\,\mathrm{kg\,d^{-1}}$) (Carabano et al., 2016). As the average milk yield of the cows in our focus regions is higher than in the mentioned studies about milk yield depression (approximately $35\,\mathrm{kg\,d^{-1}}$ to $45\,\mathrm{kg\,d^{-1}}$ milk yield), it is reasonable to scale the decline rates accordingly. In consequence for our estimation, we assumed an average decline of approximately 0.6 kg per

day and cow for each THI unit above the heat stress threshold. That means 2.4 kg less milk per day and cow for each additional day with moderate heat stress can be expected (cf. Sect. 2.6).

Our results imply an increase of on average approximately 120 heat stress days with THI $\geq 72$ by the end of the century (cf. Table 6). Eventually, the estimated milk yield losses in Germany and Spain, where our reference barns were located, accumulate to approximately 0.87% of the annual European milk yield today which is approximately $168 \cdot 10^6$ tons (cf. Tab 6).

As Germany and Spain provide together approximately 24% of the European milk yield, assuming our reference barns and the focus regions to be representative for the average change of heat stress events in Europe, the total loss can be extrapolated to be about 3.6% of the annual milk yield today under the RCP 8.5 scenario.

In addition to the milk yield, a decrease in the milk fat (approximately 0.34% to 0.4%) and protein (approximately 0.08% to 0.2%) content (with milk fat content typically around 3.5% and milk protein content typically around 2.9%) can be expected

according to literature (Bouraoui et al., 2002; Collier et al., 2012; Carabano et al., 2016). Under contemporary market conditions, higher percentages of fat and proteins increase the milk price (Bailey et al., 2005). The amount of these price corrections depends on the local markets. In addition to losing these potential bonuses, low fat and protein content increases the risk of rejection from the buyer of the milk lot.





**Table 6.** The number of heat stress events (HSE) and their average duration (HSED) based on the projections under RCP 8.5 were considered. The number of heat stress days (HSD) was estimated as the ratio HSE/HSED assuming one period of consecutive heat stress hours per day. Milk yield loss (MYL) per country was extrapolated for the countries where the reference barns are located and given relative to a reference milk yield. The assumed total amount of cows was $4.2 \cdot 10^6$ in Germany and $0.8 \cdot 10^6$ in Spain (2017 level according to de.statistica.com). For Germany half of the cows were allocated to the projected change for the reference barn Groß Kreutz and the other half to the one for the reference barn Dummerstorf. A reference annual European milk yield of $168 \cdot 10^6$ tons was assumed (cf. introduction section).

| focus region / reference barn | HSE | HSED [h] | HSD | MYL per cow [kg] | MYL per country [%] |
|---|---|---|---|---|---|
| Central Europe maritime (reference barn Dummerstorf) | 600±200 | 3±1 | 200 | 480.0 | 0.60 |
| Central Europe continental (reference barn Groß Kreutz) | 200±200 | 4±2 | 50 | 120.0 | 0.15 |
| Mediterranean (reference barn Bétera) | 1800±200 | 17±5 | 106 | 254.4 | 0.12 |

Heat stress events and thus financial losses are concentrated in the summer. An increase of 120 additional heat stress days per year, at least half of which are expected to occur in summer, implies that in each summer month approximately 20 additional heat stress days can be expected, thus affecting liquidity of farms in summer. In the worst summer month, farmers may lose approximately 17 € per cow assuming an average milk price of 35 cent per kg milk. With our estimates, a month with mild
stress would be equivalent to lose 6.6% of the monthly income. For an average farm in Germany and Spain that would involve losing 37% respectively 32% of the monthly farm gross margin without coupled payments from the year 2016. This yields particular challenges for the survival of dairy specialized farms, of which only 2% and 21% in Germany and Spain respectively have positive net economic margin (European Commission - EU FADN, 2018).

Finally, in countries with already pronounced hot summer periods, like in the Mediterranean region, where farms already
manage calving seasonally in order to avoid the lower summer fertility rates, the increase of the number of heat stress events in spring or autumn may be particularly damaging.

### 3.4.2   Environmental impact

Lower productivity per cow, as expected under heat stress (cf. Sect. 3.4.1), has been linked with increased ammonia emission intensity in the literature (Groenestein et al., 2019; Sajeev et al., 2018; Sanchis et al., 2019). The ammonia release from manure
increases with temperature by approximately 1.5 g Hence, the increase in heat stress events can be translated into the number of hours with at least 4.5 g per cow and day higher ammonia emissions. With approximately 120 more heat stress days (cf. estimation in Sect. 3.4.1) 540 g per cow and year (which is about $10^6$ mg per barn and year) higher ammonia emissions can be expected as a result of the climate change. With around $30 \cdot 10^6$ dairy cattle in Europe this would pile up to additional annual ammonia emissions from European dairy cattle husbandry of about 16 Gg per year (i.e., approximately 2.9% (550 Gg)
of the German, 4.5% (353 Gg) of the Spanish or 0.4% (3624 Gg) of the EU-28 National Emission Ceilings (NEC) target,



cf. https://www.eea.europa.eu). This implies further impacts as ammonia contributes to the formation of secondary particulate matter, which is relevant with regard to respiratory health issues and the Earth's radiation budget (Lelieveld et al., 2015). Moreover, ammonia reacts to chemical compounds that lead to acidification of soil and water (Sutton et al., 2013).

Besides ammonia, the emission of greenhouse gases, particularly methane, is a crucial topic. Although its average atmo-
spheric concentration is only a small fraction of that of carbon dioxide (1800 ppb compared to 390 ppm), methane is initially far more harmful (Pachauri et al., 2014). Methane production of ruminants is associated with microbial fermentation of hydrolyzed carbohydrates and influenced by many factors including the ambient temperature (Broucek, 2014). An optimal ambient temperature (i.e., no climatic stress) corresponds to minimal methane emissions from dairy cattle, while each heat stress event is expected to lead to a few grams per livestock unit and hour higher methane emissions (Hempel et al., 2016b). In addition, the
higher temperatures will yield a considerable increase of methane emissions from manure. The total methane increase implies a positive feedback loop as the increased methane concentration in the atmosphere will contribute to an acceleration of climate change. Based on the expected increase of the number of heat stress events, the impact of heat stress on methane emissions is about $10^6$ mg per year per dairy barn (cf. Table 7). With around $30 \cdot 10^6$ dairy cattle in Europe and assuming an average barn size of 200 cattle this corresponds to an increase of approximately 0.15 Gg/year (i.e. on average about $10^{-3}$ ppb higher
methane concentrations).

**Table 7.** The change in methane release (corresponding to annual emissions) is estimated as the product of the average livestock unit ($LU_{avg}$) in the barn and the estimated number of additional heat stress events (HSE). A 1 g $LU^{-1}$ $h^{-1}$ higher methane emission during heat stress conditions is assumed (Hempel et al., 2016b). The average livestock unit of the reference barns was estimated as number of animals in the barn times 500 kg divided by the average body weight. Considering a total mass of the earths atmosphere of approximately $5 \cdot 10^{21}$ g this methane release was converted into a concentration increase, neglecting other processes that affect the methane concentration (Trenberth and Smith, 2005).

| focus region / reference barn | $LU_{avg}$ | HSE | release [mg] | concentration increase [mg/g] | concentration increase [ppb] |
|---|---|---|---|---|---|
| Central Europe maritime | 275 | 200 | $55 \cdot 10^6$ | $1.1 \cdot 10^{-14}$ | $1.1 \cdot 10^{-8}$ |
| Central Europe continental | 40 | 600 | $24 \cdot 10^6$ | $0.5 \cdot 10^{-14}$ | $0.5 \cdot 10^{-8}$ |
| Mediterranean | 150 | 1800 | $270 \cdot 10^6$ | $5.4 \cdot 10^{-14}$ | $5.4 \cdot 10^{-8}$ |

### 3.4.3 Welfare impact

Heat stress introduces a couple of physiological impacts as cattle try to adapt themselves by increasing respiration rate (in extreme situation as panting), reducing milk yield and reproductive performance as well as by changing the feeding behavior, decreasing activity and increasing standing time (De Rensis and Scaramuzzi, 2003; West, 2003; Schütz et al., 2008; Dikmen
and Hansen, 2009). Behavior, health and productivity provide information to evaluate the welfare.

The respiration rate, measured in beats per minute (bpm), is particularly valuable to evaluate instantaneous heat stress as it is affected by the ambient conditions with little or no time lag (Pinto et al., 2019; Galán et al., 2018; Brown-Brandl et al., 2005). While under thermo-neutral conditions the respiration rate ranges from 15 bpm to 36 bpm, high-yielding dairy cattle





tend to increase their respiration rate by 27 bpm to 39 bpm if THI increases from THI $\leq$ 68 to THI $\geq$ 80 (i.e., 2-4 bpm per THI unit) (Dirksen et al., 1990; Jackson and Cockcroft, 2008; Pinto et al., 2017; Berman et al., 1985; Ominski et al., 2002). In consequence, based on our results an increase of approximately 9 bpm (i.e., 25% to 60% relative to the normal respiration rate) can be expected under RCP 8.5 during one tenth of all hours of a year or one fourth of all summer hours in addition to the

current situation in the reference barns in our focus regions. The initial response is, however, part of a homeostatic mechanism which includes besides increased respiration rate also increased water intake, increased loss of body fluids due to sweating and panting and reduction in fecal and urinary water losses, reduced feed intake, and increased heart rate during short-term exposure to heat Kadzere et al. (2002). If the heat stress persists, the muscles of the animal tend fatigue and the respiration rate tend to decrease again for a short time. Thus, with persisting heat load accumulation, the respiration rate will level off at an

intermediate value which can, however, still be considerable above the normal state. These changes promote diseases, such as disorders of the acid-base-budget (alkalosis), which increase the probability for medical treatments and in the long term may negatively affect longevity (West, 2003).

The most common behavioral indicator for heat stress is the time spent standing, where the lying posture is considered as a cow comfort indicator (Galán et al., 2018; Acatincăi et al., 2010; Herbut and Angrecka, 2018). The average daily lying time

decreases by approximately 10 to 20 minutes per THI unit under heat stress conditions resulting in an increase of the standing time in the same order of magnitude to improve the wind convection and thus increase the heat dissipation (Cook et al., 2007; Allen et al., 2015; Heinicke et al., 2018). This increased standing time is an effect of aggregated heat stress events and is typically associated with daily averaged THI. Our results imply approximately 120 additional heat stress days with prolonged periods of THI$\geq$72 (i.e. nearly 1/3 of the year). At each of those days an approximately 1 h longer standing time can be

expected. This significantly contributes to a higher risk of lameness (Cook et al., 2007; Allen et al., 2013).

The changes in the average duration of heat stress events further imply that in the Mediterranean region cattle are potentially under permanent heat stress in summer. Especially during the night, the decrease of the ambient temperature is too low for recovering (Polsky and von Keyserlingk, 2017). Hence, a behavioral adaption in terms of shifting activity towards non-heat stress hours might become impossible. Although short-term adaptation of the physiology of the cows might be supported by

the increased duration to some degree, finally, the effect of the daily heat load duration amplifies the effect of the average daily THI up to a point at which the cows can not further adapt their activity changes (Heinicke et al., 2018, 2019).

### 3.4.4   Adaptation options for animal housing

Moderate changes in the heat stress risk can be addressed by short-term measures which optimize the already implemented control mechanisms such as shading, fans, adjustable opening or cow showers and fogging devices (St-Pierre et al., 2003;

Galán et al., 2018; Davison et al., 2016; Polsky and von Keyserlingk, 2017; Honig et al., 2012; Anderson et al., 2013; Valtorta and Gallardo, 2004). Depending on the boundary conditions some of those measures may become more suitable than others.

If the annual variability of temperature is rather small and the overall heat stress risk is high, such as in Mediterranean regions, open barns can be considered as an adaptation measure. Under such conditions, as in the example of the reference barn Bétera, natural ventilation offers almost no possibility for further adaptation by adjusting the opening configurations. In





such buildings, fans and showers or fogging devices have been implemented in the past to alleviate the heat load and enable high milk yields during hot periods (Ortiz et al., 2015; Fournel et al., 2017). The fans can in principle decrease or increase the air speed in the animal occupied zone as they induce a flow that can be aligned or opposed to the naturally induced flow (Anderson et al., 2013). The speed and direction of the fan-induced flow could be optimized via velocity measurements inside

the barns. High relative humidity can additionally promote the cooling via the air flow. Similarly, showering and fogging contribute to the reduction of heat stress, particularly under dry weather conditions. If the ambient relative humidity is already high these measures are less efficient, but it has been shown that frequent showering can be still valuable (Honig et al., 2012). In terms of costs and benefits, recurring cooling sessions instead of constant cooling are reasonable, where the number of sessions is an important factor (Pinto et al., 2019). With increasing temperatures and decreasing humidity (as projected in our

study, cf. Fig. 3) frequent showering, as common for example in the Israeli husbandry system, might become more valuable for the Western Mediterranean region.

For the Central European regions, in general the cooling by showering and fogging can be expected to be less efficient than in the Mediterranean region due to the higher relative humidity. However, as hot and dry periods are expected to become more frequent, such devices can still be a valuable investment (Hübener et al.). Smart regulation of the fans and the opening

configurations will be, however, even more valuable. The position of curtains and the opening ratio have a crucial impact on the flow pattern and the air speed in the animal occupied zone. It has been shown that the average horizontal velocity in the animal occupied zone could be varied in a range of $-4\%$ up to $70\%$ of the incident flow velocity (Yi et al., 2018). This can reduce emissions, because the airflow can be controlled precisely and the overflowing of emission-active surfaces can be minimized. Moreover, the local air exchange rates in the animal occupied zones can be significantly improved.

As a mid-term adaption strategy, a sensor-based control of openings, fans and fogging devices should be implemented, including a smart control of the fogging times governed by the actual relative humidity. Active cooling may involve also the use of tubes for targeted supply of (potentially pre-cooled) air in the animal occupied zone. The speed and orientation of the mechanical ventilation support (e.g. by fans or tubes) should be regulated automatically based on wind speed measurements in the barn. In addition, further improvement of the roof insulation and the use of cooling pads in the cubicles may reduce the

thermal load. The number and timing of the cooling sessions, the operation of the fans or tubes and the control of the curtains should be based on cow-specific indicators such as respiration rate or body temperature. In this context, respiration rate is the more direct indicator, but it is much harder to measure automatically at the moment (Strutzke et al., 2018).

Finally, considering the large probability of highly increasing heat stress risk by the end of the century, the investment in hybrid ventilation systems for cattle husbandry might become valuable. An engineering solution of a precision air supply sys-

tem with additional mechanical ventilation could provide a better local thermal environment for cows and remove considerably more heat from the animals than state-of-the-art systems, especially under calm conditions (Wang et al., 2018c). Such a system may be combined with smart fogging devices and air flushing systems as well as adjustable opening configurations and a number of fans permitting to switch between natural ventilation, mechanically supported natural ventilation and mechanical ventilation in a smart, automated way.



## 4    Summary and Conclusions

Our study showed that the annual average temperature significantly increases inside the barns in our focus regions, while the relative humidity showed a decreasing trend and the wind showed no respectively a weak increasing trend. Although decreasing humidity and increasing wind speed in general alleviate the heat load, the elevated temperatures lead to a considerable increase

of the heat stress risk reflected by an increase in the number (additionally up to 21% of all hours of a year) and the duration of heat stress events (up to 17 h prolonged). The heat stress risk and the magnitude of subsequent impacts differ regionally due to different offsets and seasonal variability and depending on the assumed radiative forcing and the driving climate model. Nevertheless, considerable socio-economic and environmental impacts must be expected.

     For example, we estimated an increase of the respiration rate of up to 60% during one tenth of all hours of a year and

up to 1 h prolonged standing times at one third of the days, which promotes health issues and increases the probability of medical treatments. This implies additional costs for the farmers as well as potential secondary impacts on human health (e.g., increased bioburden of barns as animals become more susceptible for infectious diseases or further spreading of antibiotica resistances). At the same time, milk yield was estimated to decrease by about 3.5% relative to the present European milk yield. In consequence, farmers may expect financial losses particularly during the summer season of about 6.6% of the monthly

income. In addition, an increasing demand for emission reduction measures must be expected. We estimated that an emission increase of about 16 Gg ammonia per year and 0.1 Gg methane per year can be expected, implying feedback loops in the climate system which are presently underexplored.

     The multiple impacts highlight the urgent need for an adaptation of the husbandry system. The most common approach is the adaptation of housing. For example, short-term measures which optimize the already implemented control mechanisms in dairy

cattle buildings such as fans, air flushing systems with active cooling, adjustable openings, cow showers and fogging devices can be valuable. Furthermore, smart and automated control systems based on animal-associated sensors can substantially increase the efficiency of those devices and alleviate the impacts.

     Our impact assessment demonstrated the diversity and complexity of climate change impacts on dairy cattle. We estimated the order of magnitude of potential impacts of heat stress in the European dairy cattle husbandry system. However, the presented

impacts were estimated based on a variety of simplifications implying that further quantitative studies on the direct and indirect economic, environmental and ethic (i.e. animal welfare aspects) impacts are required. This includes, for example, research on the dependency of ammonia, methane and other pollutant emissions on the ambient conditions (such as air temperature) as well as the building design and management. Moreover, the relation between heat stress induced physiological and behavioral changes, health issues and medical treatments must be investigated in more detail. In addition, scenarios for future adaptation

(e.g., breeding, housing, feeding, acclimatization) and future milk yields and milk prices should be further developed and included in forthcoming studies. The same applies to the accumulation of heat stress impacts with increasing duration of heat load and differentiated by heat stress levels (e.g., mild, moderate, severe) and breed.

     Our heat stress projections and the subsequent impact assessment were based on several models that involved different levels of uncertainty. The uncertainty analysis revealed knowledge gaps that require further detailed research in the future involving,



for example, studies about the refinement of regional climate model projections, particularly with regard to changes in the wind regime. Moreover, feedback loops related to the radiative forcing which are implied by an emission increase with temperature rise should be investigated in-depth. This includes, for example, the accelerated increase of atmospheric methane concentration or ammonia-induced formation of particulate matter. It also involves detailed research on the accuracy and representativeness

5 of measurements of agricultural emissions. Furthermore, in-depth understanding of animal-individual heat stress thresholds dependent on activity and vitality, posture and position instead of herd and barn averaged indices and thresholds is required.

Finally, future research must incorporate the further development and refinement of indoor climate models. This applies to the statistical as well as to dynamical approaches such as Raynolds-averaged Navier-Stokes (RANS), Large Eddy Simulation (LES) and Direct Numerical Simulation (DNS) approaches for naturally ventilated barns. The applicability of statistical models

10 under climate change conditions is per se limited. Hence, a generalization and extrapolation to other regions and climate zones including cross-validation with further measurement data and simulations of dynamical indoor models would be valuable to evaluate possible adaptation and mitigation strategies in future studies.

*Data availability.* Hempel, Sabrina; Menz, Christoph (2019), "Indoor climate projections for European cattle barns", Mendeley Data, v1, http://dx.doi.org/10.17632/tjp8h523p7.1



## Appendix A: Table of Nomenclature and Abbreviations

**Table A1.** Nomenclature and abbreviations used in this manuscript.

| | |
|---|---|
| GCM | global climate model – general circulation model to describe climate behavior by integrating a variety of fluiddynamical and chemical equations that are derived directly from physical laws or constructed by empirical means |
| RCM | regional climate model – numeric climate prediction model forced by specified lateral and ocean conditions from a GCM or observation-based dataset that accounts for high-resolution topographical data, land-sea contrasts and surface characteristics |
| RCP | representative concentration pathway – greenhouse gas concentration trajectory which describe climate futures with a particular radiative forcing values in the year 2100 |
| ReLU | rectified linear unit – activation function in neuronal networks defined as the positive part of its argument |
| THI | temperature humidity index – empirical model to evaluate the thermal environment as a function of air temperature and humidity |
| ETIC | equivalent temperature index for cattle – empirical model to evaluate the thermal environment as a function of air temperature, humidity, air speed and radiation |
| HSE | heat stress events – number of consecutive hours of at least moderate heat stress as defined by a THI- or ETIC-threshold |
| HSED | heat stress event duration – length of periods of consecutive hours of at least moderate heat stress |
| HSE | number of heat stress days – amount of days with hours of at least moderate heat stress (approximated by HSE/HSED) |
| NEC | national emission ceilings – European directive which sets national reduction commitments for pollutants which lead to significant negative impacts on human health and the environment |

*Author contributions.*  SH made major contributions to the general setup of the study, the conception of the manuscript, the data analysis and the writing process (original draft, review and editing) in all sections and contributed to the on-farm measurements and data curation.

CM made major contributions to the general setup of the study, the conception of the manuscript, the data curation, the model development and the visualization and contributed and to the writing process (original draft, review and editing) in all sections.

SP contributed to the general setup of the study, the on-farm measurements and the writing process (original draft, review and editing) with major contributions on stress indices and welfare impacts.

EG contributed to the general setup of the study and the writing process (original draft, review and editing) with contributions to welfare impacts and adaptation and major contributions on economic impacts.

DJ contributed to the general setup of the study, the on-farm measurements, the data curation and the writing process (original draft, review and editing) with major contributions to adaptation.

FE contributed to the general setup of the study, the on-farm measurements and the writing process (review and editing) with a focus on environmental impacts.

TMS contributed to the general setup of the study, the on-farm measurements and the writing process (review and editing) with a focus on stress indices and welfare impacts.

XW contributed to the writing process (review and editing) with a focus on methods and adaptation.





JH contributed to the on-farm measurements and the writing process (review and editing) with a focus on stress indices and welfare.

GZ contributed to the general setup of the study and the writing process (review and editing) with a focus on adaptation.

BA contributed to the general setup of the study and the writing process (review and editing) with a focus on the impacts.

AdP contributed to the general setup of the study and the writing process (review and editing) with a focus on economics.

TA contributed to the general setup of the study and the writing process (review and editing).

*Competing interests.* The authors declare that there are on competing interests.

*Acknowledgements.* This study was conducted in the framework of the OptiBarn project in the FACCE ERANET+ initiative on climate smart agriculture. The research was financially supported by:

– the German Federal Ministry of Food and Agriculture (BMEL) through the Federal Office for Agriculture and Food (BLE), grant

numbers 2814ERA02C and 2814ERA03C

– the Instituto Nacional de Investigación Tecnología Agraria y Alimentaria (INIA) grant number 618105

– the Basque Government through the BERC 2018-2021 program

– the Spanish Ministry of Economy and Competitiveness MINECO through BC3 María de Maeztu excellence accreditation MDM-2017-0714

– the Innovation Foundation Denmark, DK-Grant No. 4215-00004B

Elena Galán is financed by the the grant Juan de la Cierva 2016 (FJCI-2016-30263) from the Spanish Ministry of Economy, Industry and Competitiveness. Agustin del Prado is financed by the programme Ramon y Cajal from the Spanish Ministry of Economy, Industry and Competitiveness (RYC-2017-22143).

We thank Klaus Parr, Detlef May and Ramón Morla for permitting to conduct the on-farm measurements and Carola Franke (LVAT) for

her assistance in managing / scheduling the intensive on-farm measurements in the barn Groß Kreutz.

We further thank Knut Schröter, Ulrich Stollberg and Andreas Reinhardt, technicians at ATB, for supporting the implementation of the on-farm measurements.

We further thank the ReKliEs-De project (funded by German Ministry of Education and Research, grant 01LK1401) team for provide us with the RCM simulations used within this study.



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
