# Peer review of "Heat stress risk in European dairy cattle husbandry under different climate change scenarios - uncertainties and potential impacts"

_Earth System Dynamics, 2019_

## Referee Comment (RC1) · Grant Dewell (Referee) · 20 Jun 2019

Interesting manuscript evaluating climate change on heat stress impacts in European dairy cattle. Model development is appropriate and authors have acknowledged potential shortcomings. Paper is generally well written. I am not familiar with this journal but text was more conversational then most scientific journals I read. Generally try not to use we did this or our analysis.

Specific comments

Page 1, line 17, not sure what sentence means, especially "region respectively the

barn"

Page 2, line 24, is there a reference for this or is this opinion? If opinion change considered to believed. I would argue that livestock are not more efficient and genetic adaptation is slower.

Page 3, line 10-12, this sentence is self-defeating. The fact that economic drivers are not triggering mitigation strategies may be negating premise of paper.

Page 3, line 30, Adaption and recovery sentence seems unnecessary. I know you discuss recovery later in paper but this sentence seems like an afterthought here. Either expand on importance or delete.

Page 4, line 6, human health implication is a stretch, particularly without a reference.

Page 4, line 19, don't think you need this last sentence.

Page 4, line 22-32, very conversational, lot of we and our. Definitely delete Eventually in line 29

Page 12, line 20-24, very conversational again. I am not used to a having an introductory paragraph like this in front of each section, check if it is appropriate for journal

Page 18, line5, I would delete this last sentence

Page 23, line 2-4, I like this intro better than the conversational intros in some of earlier sections.

Page 23, line 19, need to justify the 0.6 kg better a 50% inflation over what was reported in other studies seems like a pretty steep scale or drop it down a little if just a guess.

Page 24, line 13, I think you are over interpreting what has been reported in literature. Lower productive cows are less efficient and may have increased ammonia emission compared to high producing cows. Heat stress doesn't change the cow's genetic efficiency, it impacts their behavior and thereby their feed intake which decreases

productivity but if they are eating less they are producing less ammonia.

Page 25, line 10, need reference for increase in methane from manure. Think you are implying that at higher temperatures more methane is released from manure due specifically to methane volatility not to amount of methane in manure. Is this relevant? Pile of composting manure is hotter than ambient temperature so is ambient temperature relevant to methane release?

Page 26, line 26, may be useful here to add short discussion about Middle-Eastern and Tropical dairies. You are kind of concluding that Mediterranean region will be too hot to allow for dairy industry but they have dairy cattle in hotter environments already.

Page 28, line 11-12, think you are stretching human health impact. Most of health issues see in heat stress are metabolic in nature and therefore no need for antibiotics.

Technical corrections

Page 1, line 1, delete exceptional. There is a warming trend not sure we can call it exceptional

Page 1, line 5, delete however

Page 1, line 6, delete Moreover

Page 2, line 17, change was to is expected to be,

Page 3, line 31, delete The

Page 17, line 3, diverse is misspelled, also delete While for at beginning of next sentence

Page 25, line 21, breaths per minute not beats.

Page 26, line 8, muscles of the animal tend TO fatigue

---

## Referee Comment (RC2) · 15 Aug 2019

Dear Svenja Lange We accept for publication that attached paper with some notices in revised paper With respect you Prof. Dr. alsaied alnaimy habeeb

Please also note the supplement to this comment:
https://www.earth-syst-dynam-discuss.net/esd-2019-15/esd-2019-15-RC2-supplement.pdf
* * *
[Figure]

**Heat stress risk in European dairy cattle husbandry under different climate change scenarios - uncertainties and potential impacts**

Sabrina Hempel[1], Christoph Menz[2], Severino Pinto[1], Elena Galán[3], David Janke[1], Fernando Estellés[4], Theresa Müschner-Siemens[1], Xiaoshuai Wang[5], Julia Heinicke[1], Guoqiang Zhang[5], Barbara Amon[1], Agustín del Prado[3,6], and Thomas Amon[1,7]

[1]Leibniz Institute for Agricultural Engineering and Bioeconomy (ATB), Max-Eyth-Allee 100, 14469 Potsdam, Germany
[2]Potsdam Institute for Climate Impact Research (PIK), Telegraphenberg A 31, 14473 Potsdam, Germany
[3]Basque Centre for Climate Change (BC3), Sede Building 1, 1st floor, Scientific Campus of the University of the Basque Country, 48940 Leioa, Spain
[4]Institute of Animal Science and Technology, Universitat Politècnica de València, (UPV), Camino de Vera, s/n 46022 Valencia, Spain
[5]Aarhus University (AU), Department of Engineering, Blichers Allé 20, P.O. Box 50, 8830 Tjele, Denmark
[6]Basque Center for Applied Mathematics (3BCAM), Alameda de Mazarredo 14, 48009 Bilbao, Bizkaia
[7]Free University Berlin (FUB), Department of Veterinary Medicine, Institute of Animal Hygiene and Environmental Health

**Correspondence:** Sabrina Hempel (shempel@atb-potsdam.de)
**Abstract.** In the last decades, an exceptional global warming trend was observed. Along with the temperature increase, modifications in the humidity and wind regime amplify the regional and local impacts on livestock husbandry. Direct impacts include

---

## Author Comment (AC1) · 20 Aug 2019

**1   ESDD – Review 1**

Interesting manuscript evaluating climate change on heat stress impacts in European dairy cattle. Model development is appropriate and authors have acknowledged potential shortcomings. Paper is generally well written. I am not familiar with this journal but text was more conversational then most scientific journals I read. Generally try not to use we did this or our analysis.

We thank the reviewer for his valuable comments. As the topic is very interdisciplinary, we tried to avoid the use of too many discipline-specific terms. This may let the text appear rather conversational in some parts. When processing the specific comments, we pay particular attention to minimize the use of phrasings like "we did" and "our analysis".

**1.1 Specific comments**

Page 1, line 17, not sure what sentence means, especially "region respectively the barn"

This sentence was intended to emphasise that the investigated barns are located in different (climatic) regions. Barn design / management affect the reaction to local climatic conditions. In addition, different climate models and RCP scenarios project different local climatic conditions. All together affects the heat stress risk projection. We rephrased the sentence to make our point more clear: "The impacts of the projected increase of heat stress risk varied among the barns due to different location and design as well as the anticipated climate change (considering different climate models and future greenhouse gas concentrations). There was an overall increasing trend in number and duration of heat stress events."

Page 2, line 24, is there a reference for this or is this opinion? If opinion change considered to believed. I would argue that livestock are not more efficient and genetic adaptation is slower.

The remark is based on some review papers, but it is related to (1) intensive livestock farming and (2) direct physiological adaption, not genetic adaptation. We regret that we haven't made our point clear here. As this is rather an afterthought, we will refrain from a more detailed explanation in the text and remove the remark.

Page 3, line 10-12, this sentence is self-defeating. The fact that economic drivers are

not triggering mitigation strategies may be negating premise of paper.

We apologize that our phrasing was ambiguous in this paragraph. It was not our intention to say that economic drivers are not triggering mitigation strategies. We rather intended to highlight that climate change adaptation and heat stress mitigation are (too) little considered in breeding strategies so far compared to other economic drivers such as high production rates. Adaption and mitigation via modification in housing on the other hand has gained some interest also in the scientific community. Measures and systems are, however, not optimized due to a lack of understanding of the complex livestock environment interaction which involves diverse factors and feedback loops. We rephrased the paragraph: "Breeding is one possibility to reduce the impacts of climatic stress (Hammami et al., 2014). However, climate change is a slow process, feedback mechanisms are not fully understood and there are contradictory aims (i.e. low heat stress susceptibility versus high yields) (Hoffmann, 2010). In consequence, climate change adaptation or heat stress mitigation, respectively, play only a minor role in breeding strategies. Modifications in housing management are the main measures taken to improve the ability of livestock to cope with climatic stress conditions."

Page 3, line 30, Adaption and recovery sentence seems unnecessary. I know you discuss recovery later in paper but this sentence seems like an afterthought here. Either expand on importance or delete.

We agree that for the introduction the recovery aspect is rather an afterthought. We deleted the sentence.

Page 4, line 6, human health implication is a stretch, particularly without a reference.

We agree that potential secondary implications on human health are not in the focus of this study. As a detailed discussion of those impacts would be out of the scope of this paper, we removed the remark on human health from the introduction.

Page 4, line 19, don't think you need this last sentence.
We removed the sentence.

Page 4, line 22-32, very conversational, lot of we and our. Definitely delete Eventually in line 29.

These paragraphs are rephrased avoiding "we did" or "our analysis" as suggested. We removed "eventually".

Page 12, line 20-24, very conversational again. I am not used to a having an introductory paragraph like this in front of each section, check if it is appropriate for journal

We checked the journal guidelines for authors and several papers which were published in this journal and haven't found a standard procedure with regard to introductory paragraphs. There are papers with such an introductory paragraph and papers without it. We came to the conclusion that both options are appropriate for a publication in the journal. We agree, however, that some parts of our introductory paragraph are a repetition of what was said earlier. We rephrased and shortened the paragraph: "For the sake of simplicity, projections of the indoor climate and the estimated heat stress risk for the housed dairy cattle are shown for individual barns. Seasonal characteristics as well as differences between the three RCP scenarios and the two stress indices emerge."

Page 18, line5, I would delete this last sentence

We deleted the sentence.

Page 23, line 2-4, I like this intro better than the conversational intros in some of earlier sections.

We used the format of this intro as some kind of template for the rephrasing of the other introductory paragraphs.

Page 23, line 19, need to justify the 0.6 kg better a 50% inflation over what was reported in other studies seems like a pretty steep scale or drop it down a little if just a guess.

We agree that up to 50% inflation appears pretty steep at the first glance. The reasoning behind is the following: The herds reported in the studies of Bouraoui et al. and Bohmanova et al. we consider as "average producing cows" (20 -30 kg daily milk yield). In the study of Carabonao et al. the most productive cows (> 40 kg daily milk yield) lost 0.174 kg/day per THI unit more than the average producing cows (around 30 kg daily milk yield). Taking the values from Bouraoui et al. and Bohmanova et al. and adding the value from Carabonao et al., we end up with a range of 0.474 kg/day to 0.584 kg/day for highly productive cows as they are common in our focus barns. We decided to consider a worst case scenario (value of the upper bound of that range) without genetic adaptation (e.g. strong and fast temperature rise) and taking into account that the average producing cows in the studies of Bouraoui et al. and Bohmanova et al. were even less productive than the average cows in the study of Carabanao et al.

Following the reviewers suggestion we adapted our estimation using now a value in the middle of the range (i.e. a decrease of 0.5 kg/day instead of 0.6 kg/day). By that our extrapolation ends up with milk yield losses of 0.68% instead of 0.87% of the milk yield in Germany and Spain, or 2.8% instead of 3.6% of the annual European milk yield. Extrapolated losses for farmers will be 14 Euro (5.4% of monthly income) instead of 17 Euro (6.6% of monthly income), i.e. 30% instead of 37% (Germany) or 26% instead of 32% (Spain) of monthly farm gross margin.

Page 24, line 13, I think you are over interpreting what has been reported in literature. Lower productive cows are less efficient and may have increased ammonia emission compared to high producing cows. Heat stress doesn't change the cow's genetic efficiency, it impacts their behavior and thereby their feed intake which decreases productivity but if they are eating less they are producing less ammonia.

We agree that heat stress doesn't change the genetic efficiency, but mainly affects feed intake which changes the composition of excrements and urine and in consequence the composition of manure and slurry. This implies a reduction of ammonia emissions. On the other hand, heat stress conditions are associated with high temperature. Following Arrhenius law this significantly speeds up reaction kinetics and in consequence leads to higher ammonia emissions. Although those two competing effects exist, measurements in cattle barns reported in early studies clearly show that in a temperature range of approximately 10°C to 40°C the reaction kinetics are the dominant driving force and emissions are in general increasing (cf. for example Hempel et al., 2016 or Sanchis et al., 2019 from the reference list in our manuscript).

Page 25, line 10, need reference for increase in methane from manure. Think you are implying that at higher temperatures more methane is released from manure due specifically to methane volatility not to amount of methane in manure. Is this relevant? Pile of composting manure is hotter than ambient temperature so is ambient temperature relevant to methane release?

We added a reference for the increase in methane from manure. Methane emissions from manure management are heavily dependent on ambient temperature (see e.g. IPCC Guidelines for National Greenhouse Gas Inventories, volume 4 (agriculture) chap. 10 (emissions from livestock and manure management; or Amon, B; Kryvoruchko, V; Frohlich, M; Amon, T; Pollinger, A; Mosenbacher, I; Hausleitner, A, (2007): Ammonia and greenhouse gas emissions from a straw flow system for fattening pigs: Housing and manure storage. LIVEST SCI. LIVESTOCK SCIENCE; 112: 199-207). It is not the volatility that increases, but the methane production conducted by anaerobic bacteria, which strongly dependents on ambient temperature. This effect is utilized e.g. in biogas plants where the digesters are heated to increase methane production. Our estimations are based on dairy housing systems with liquid manure, not on composting systems (the latter are less temperature dependent than slurry systems.) We added a remark on that point. In contrast to anerobic liquid manure systems, in aerobic systems such as composting methane formation processes are generally of minor importance.

Page 26, line 26, may be useful here to add short discussion about Middle-Eastern and Tropical dairies. You are kind of concluding that Mediterranean region will be too

hot to allow for dairy industry but they have dairy cattle in hotter environments already.

It was not our intention to convey that the Mediterranean region will be too hot for dairy industry by the end of the century, but rather to highlight that adaptation (of breeds and husbandry system) is urgently needed. We apologize to be not clear in that point. We added the following short discussion at the end of the subsection: "In consequence milk yield is expected to decrease significantly if no additional cooling is provided. It has to be noted that the dairy husbandry in the Mediterranean region, as well as in countries in Middle-Eastern or in tropical regions, is already faced with extended periods of heat load conditions along the year (Honig et al., 2012, Costa et al., 2015, Ortiz et al., 2015). The associated reactions to heat stress conditions have a strong genetic component (Broucek et al. (2007), Bernabucci et al. (2014)). If climate changes sufficiently slow, dairy herds will genetically adapt by nature to some degree to the elevated temperatures. In addition, most of the hot countries already search actively for adaptation measures to alleviate the cows' heat stress (cf. Sec. 3.4.4). Those measures include, for example, promoting cross breeds adapted to the heat load conditions (Costa et al., 2015), and evaporative cooling systems to provide refreshment for the cows, especially during the day when the environment temperature is particularly high (Honig et al., 2012, Ortiz et al., 2015, Pinto et al., 2019b, Berman, 2006, Broucek et al., 2007, Avendano-Reyes et al., 2010, Legrand et al., 2011, Calegari et al., 2016). These efforts need to be further intensified in the future."

Page 28, line 11-12, think you are stretching human health impact. Most of health issues see in heat stress are metabolic in nature and therefore no need for antibiotics.

We agree that potential implications on antibiotics are indirect. There are recent studies showing that with temperature rise the growth of bacteria and viruses is amplified which increases the probability for infections. Potential secondary impacts on human health can be expected since animals under heat stress that suffer from metabolic illness can be expected to be more susceptible also to bacterial and viral infections. However, these aspects are not investigated in detail in our study. We removed the remark.

[Figure]

**1.2 Technical corrections**

Page 1, line 1, delete exceptional. There is a warming trend not sure we can call it exceptional

We agree that if the trend can be call exceptional or not is very much depended on the considered time scale. We deleted "exceptional".

Page 1, line 5, delete however

We deleted "however".

Page 1, line 6, delete Moreover

We deleted "moreover".

Page 2, line 17, change was to is expected to be,

We changed "was" to "is expected to be".

Page 3, line 31, delete The

We deleted "the".

Page 17, line 3, diverse is misspelled, also delete While for at beginning of next sentence

We changed "divers" to "diverse".

Page 25, line 21, breaths per minute not beats.

We changed "beats" to "breaths".

Page 26, line 8, muscles of the animal tend TO fatigue.

We added "to".

**Supplement:**

[revised manuscript text omitted]

---

## Author Comment (AC2) · 20 Aug 2019

**1  ESDD – Review 2**

We accept for publication that attached paper with some notices in revised paper. Please also note the supplement to this comment.

We thank the reviewer for the valuable comments and made the suggested formal amendments (cf. also supplement to the reply to review 1).